# L2G: Repurposing Language Models for Genomics Tasks

**Wenduo Cheng**                                          *wenduoco@andrew.cmu.edu*
*Ray and Stephanie Lane Computational Biology Department*
*Carnegie Mellon University*

**Junhong Shen**                                          *junhongs@andrew.cmu.edu*
*Machine Learning Department*
*Carnegie Mellon University*

**Mikhail Khodak**                                          *mkhodak@princeton.edu*
*Princeton Language & Intelligence, Princeton AI Lab*
*Princeton Language*

**Jian Ma**                                          *jianma@cs.cmu.edu*
*Ray and Stephanie Lane Computational Biology Department*
*Carnegie Mellon University*

**Ameet Talwalkar**                                          *talwalkar@cmu.edu*
*Machine Learning Department*
*Carnegie Mellon University*

**Reviewed on OpenReview:** *https://openreview.net/forum?id=5NM4guc90N*

## Abstract

Pre-trained language models have transformed the field of natural language processing (NLP), and their success has inspired efforts in genomics to develop domain-specific foundation models (FMs). However, creating high-quality genomic FMs from scratch is resource-intensive, requiring significant computational power and high-quality pre-training data. The success of large language models (LLMs) in NLP has largely been driven by industrial-scale efforts leveraging vast, diverse corpora and massive computing infrastructure. In this work, we aim to bypass the data and computational bottlenecks of creating genomic FMs from scratch and instead propose repurposing existing LLMs for genomics tasks. Inspired by the recently observed 'cross-modal transfer' phenomenon – where transformers pre-trained on natural language can generalize to other modalities – we introduce L2G, which adapts a pre-trained LLM architecture for genomics using neural architecture search and a novel three-stage training procedure. Remarkably, without requiring extensive pre-training on DNA sequence data, L2G achieves superior performance to fine-tuned genomic FMs and task-specific models on more than half of tasks across multiple genomics benchmarks. In an enhancer activity prediction task, L2G further demonstrates its capacity to identify significant transcription factor motifs. Our work not only highlights the generalizability and efficacy of language models in out-of-domain tasks such as genomics, but also opens new avenues for more efficient and less resource-intensive methodologies in genomic research.

## 1 Introduction

In recent years, large-scale pre-trained models, often referred to as foundation models (FMs) (Bommasani et al., 2021), have revolutionized the field of natural language processing (NLP). Models such as BERT (Devlin et al., 2019), LLaMA (Touvron et al., 2023), and GPT (Brown et al., 2020) leverage self-supervised pre-training on vast amounts of unlabeled text data to develop a rich and nuanced understanding of language. Similarly, DNA sequences can be viewed as strings of nucleotides, with recurring patterns analogous

to reusable elements in natural language. This parallel has inspired the development of genomic FMs, such as DNABERT (Ji et al., 2021), Nucleotide Transformer (Dalla-Torre et al., 2023), and HyenaDNA (Nguyen et al., 2024), which are pre-trained on large-scale genomic sequence data with subsequent fine-tuning. These models have shown potential in predicting complex genomic features, such as regulatory elements, chromatin accessibility, and gene expression.

However, constructing these domain-specific FMs from scratch is costly and resource-intensive. For instance, training the Nucleotide Transformer required approximately 174 billion tokens and 28 days of continuous training on 128 NVIDIA A100 GPUs. Even smaller models like DNABERT-2 required two weeks of training on 8 GTX 2080 Ti GPUs (Zhou et al., 2023b). More recent architectures with fewer parameters than transformer-based models, such as HyenaDNA (Nguyen et al., 2024) and Mamba (Gu & Dao, 2023), still require substantial compute budget and massive training corpora (see **Fig.** 1B and **Table** 1 for details).

To address these challenges, we explore an alternative strategy to bypass genomic pre-training altogether. Our framework, Language-to-Genome (L2G), adapts pre-trained language models to genomic prediction tasks. This work is motivated by recent advances in the *cross-modal transfer* paradigm, which leverages the general reasoning capacity of pre-trained LLMs in domains such as protein property prediction (Vinod et al., 2023) and solving partial differential equations (Shen et al., 2024a). These studies demonstrate that transferring existing models from well-studied text and vision modalities to scientific applications holds the promise of not only drastically reducing the required computational and data resources associated with pre-training, but also improving downstream model performance. L2G builds on a general-purpose cross-modal transfer approach (Shen et al., 2023) but incorporates neural architecture search (NAS) and a novel three-stage training procedure (**Fig.** 2) to adapt to the unique features of genomics data, significantly enhancing empirical effectiveness. **Fig.** 1A contrasts L2G with the traditional approach of building genomic FMs.

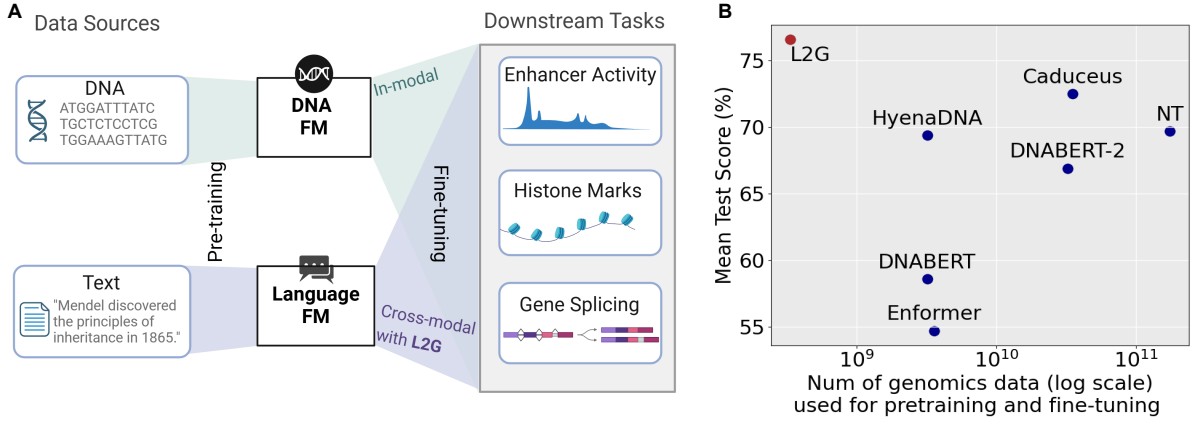

Figure 1: L2G is a data-efficient cross-modal fine-tuning method for genomics. **A**. Schematic overview contrasting genomic foundation models (FMs) with L2G during pre-training and fine-tuning. Genomic FMs pre-train with massive DNA sequencing data, while L2G bypasses genomic-specific pre-training altogether by leveraging existing pre-trained language models. Both approaches perform fine-tuning for specific downstream genomic task, but L2G's three-stage cross-modal fine-tuning workflow on par with vanilla fine-tuning in terms of compute and data requirements. **B**. L2G (red dot) achieves a higher mean test score than leading genomic FMs on the Nucleotide Transformer benchmark, with higher values indicating better performance. By skipping pre-training, L2G requires significantly less genomic data and computational resources.

The advantages of L2G are three-fold. First, by sidestepping pre-training, L2G is much more data and compute efficient. For a given downstream task, our three-stage workflow is comparable to vanilla fine-tuning in terms of required compute and data resources. All our experiments can be performed on a single A6000 GPU in a matter of hours by leveraging existing open-source language models, compared to days of training needed to develop genomic FMs from scratch. Second, L2G demonstrates better average performance than

fine-tuned genomic FMs on various genomics benchmarks (**Fig.** 1B), including GenomicBenchmarks (Grešová et al., 2023) and NucleotideTransformerBenchmarks (Dalla-Torre et al., 2023). Third, we show that L2G can tackle challenging regulatory activity prediction tasks, such as predicting developmental and housekeeping enhancer activity, where it consistently outperforms expert-designed models. Additionally, L2G learns relevant transcription factor motifs.

Overall, L2G leverages pre-trained LLMs for genomic prediction, achieving performance competitive with in-modal transfer on various genomic tasks while bypassing the massive costs associated with collecting and processing large amounts of unsupervised genomic sequencing data.

## 2 Methods

### 2.1 Cross-modal fine-tuning

Fine-tuning has been a highly effective technique for adapting pre-trained language models to various downstream tasks. However, most existing research focuses on *in-modal* adaptation, where the fine-tuning data originates from the same modality as the pre-training data but is tailored for a more specialized focus, such as sentiment analysis or text classification. In such cases, the model operates on the same type of input it was originally trained on. In contrast, *cross-modal* fine-tuning adapts a pre-trained model to work with data from unseen modalities, such as using text-pre-trained LLMs to address biological questions. **Fig.** 1A illustrates the distinction between in-modal and cross-modal fine-tuning strategies for genomic tasks.

Cross-modal fine-tuning is more challenging than the in-modal fine-tuning due to the modality gap between the pre-training and target task data (Lu et al., 2022). Bridging this gap often requires additional data alignment. For instance, to repurpose a pre-trained BERT model for predicting physicochemical and biomedical properties of protein sequences, Vinod et al. (Vinod et al., 2023) introduced R2DL (Representation Reprogramming via Dictionary Learning), a token-level alignment method that learns a sparse linear mapping between English vocabulary embeddings and amino acid embeddings.

Recently, Shen et al. (2023) proposed a more general distributional alignment technique, ORCA, that adapts various pre-trained transformer models to diverse non-text, non-vision inputs. ORCA employs a convolutional neural network to transform input data into sequence features, minimizing the distribution distance between target data embeddings and standard English token embeddings prior to fine-tuning. ORCA achieves state-of-the-art results on three benchmarks containing over 60 datasets from 12 modalities, outperforming a wide range of general-purpose, automated machine learning (AutoML) and task-specific methods. However, genomics is one domain where ORCA does not show superior results (Shen et al., 2023). Specifically, on DeepSEA – a well-known dataset for predicting functional effects of genomic sequence – ORCA falls behind non-pre-trained AutoML baselines. This motivates us to study how cross-modal alignment can be improved specifically for genomic prediction tasks.

Beyond ORCA, several other studies have also proposed different cross-modal fine-tuning strategies (Cai et al., 2024; Ma et al., 2024; Shen et al., 2024b; Zhou et al., 2023a; Chang et al., 2024; Roberts et al., 2023). However, they are not tailored to biological domains. To address this gap, we aim to develop the first cross-modal fine-tuning framework specifically designed for genomic applications.

### 2.2 Motivation for the framework

Since ORCA is the only previous method that has attempted genomics tasks, we thoroughly examine its workflow to identify limitations. Specifically, ORCA first creates custom embedder and predictor networks to support various tasks. The embedder is trained to minimize the optimal transport dataset distance (OTDD) between a target and proxy dataset, aiming to map the target dataset into the embedding space of the pre-trained model. Finally, the entire model – comprising the embedder, transformer, and predictor – is fully fine-tuned on the target task data to adapt the pre-trained model to the target modality.

However, ORCA has two major limitations when applied to genomics – both model-wise and training-wise. Model-wise, ORCA employs a universal CNN structure as the input embedder for all tasks. This embedder,

consisting of a single-layer CNN with small kernel sizes and strides designed for computer vision, may not be well-suited for genomics tasks. It cannot effectively model the long-range dependencies of genomic sequences and may fail to capture important features from genomic datasets. To address this, we propose a redesigned embedder architecture tailored for genomics data.

Training-wise, by reproducing ORCA experiments on DeepSEA, we revealed that a lower alignment loss at the end of embedder training does not necessarily lead to better downstream performance on genomics tasks. This suggests a more complex dynamic between embedder training and fine-tuning than the ORCA paper indicated. For instance, in many cases, training the embedder for longer epochs can *hurt* the final performance on the target task. We hypothesize that this occurs because a single alignment loss is insufficient for effective embedder training; closer mapping to text embeddings can result in the loss of important class information in the target genomic dataset. To address this, we propose a new embedder training objective that jointly optimizes for both distribution alignment and downstream task performance. Another possible factor is overfitting, as ORCA relied on a relatively small and imbalanced source dataset during the alignment step. To mitigate this, we sampled additional data points from the source dataset, ensuring they were evenly distributed across different categories.

By tailoring the embedder architecture and objective design to genomics data, we significantly improve empirical performance and develop a new cross-modal fine-tuning workflow, named L2G, to effectively adapt language models for genomic applications.

### 2.3 Model design

#### 2.3.1 Problem Setup

A modality $M$ consists of a feature space $\mathcal{X}$, a label space $\mathcal{Y}$, and a joint probability distribution $P(\mathcal{X}, \mathcal{Y})$. We focus on the cross-modal setting in this paper. That is, the target genomics modality $M_t$ and source language modality $M_s$ have different feature spaces, label spaces, and joint probability distributions, i.e., $\mathcal{X}^t \neq \mathcal{X}^s$, $\mathcal{Y}^t \neq \mathcal{Y}^s$, and $P(\mathcal{X}^s, \mathcal{Y}^s) \neq P(\mathcal{X}^t, \mathcal{Y}^t)$. Our goal is to adapt a model pre-trained in $M_s$ to the tasks in $M_t$.

Following previous work (Cai et al., 2024; Shen et al., 2023), the model architecture of L2G is composed of three parts: a CNN embedder, a transformer encoder, and a linear predictor (**Fig.** 2A). The embedder maps input genomics data to an embedding space, the encoder extracts features from these embeddings, and the predictor maps the encoder output to the label space.

#### 2.3.2 Embedder

Denote $f^s$ as the source embedder of a language model, which transforms the source raw data $\mathcal{X}^s$ into source language embeddings $h^{\text{text}} = R^{N \times D}$, where $N$ denotes the embedding length and $D$ denotes the embedding dimension. Following ORCA, we use the CoNLL-2003 dataset (Sang & De Meulder, 2003) as the reference dataset for text. This dataset contains nine classes for a named entity recognition (NER) task, from which we sampled 350 data points per class, resulting in a total of 3,150 data points. The source language embedder is taken from a pre-trained language transformer and remains frozen during the entire training process. Reference data is passed through the embedder to obtain reference embeddings for alignment.

Denote $f^t$ as the custom target embedder, which transforms the target genomics sequence data in $\mathcal{X}^t$ into target embeddings $h^{\text{DNA}} = R^{N \times D}$. The key to cross-modal transfer is to learn $f^t$ so that it maps $h^{\text{DNA}}$ into the shared representation space with $h^{\text{text}}$. As mentioned above, existing work typically uses a generic small-kernel convolutional layer for $f^t$, which is unsuitable for modeling long-sequence genomics data. To address this, we propose using a larger, more capable dilated CNN as the backbone architecture for $f^t$.

We chose CNN as the backbone for DNA embeddings because CNNs are well-suited for DNA embeddings as evident in previous work. Landmark models like DeepBind (Alipanahi et al., 2015) and DeepSEA (Zhou & Troyanskaya, 2015) used shallow CNNs to predict protein binding and chromatin features, and they remain standard baselines in computational genomics. More recent architectures (Kelley et al., 2018; Fudenberg et al., 2020; Avsec et al., 2021b; Yang & Ma, 2022) build on this foundation by incorporating dilated

convolutions or residual connections to capture longer-range dependencies, but the core use of convolutions remains central. CNNs excel at learning local patterns, and in genomic data, they can effectively identify short regulatory patterns, such as transcription factor binding motifs, splice signals, and repeat elements, regardless of their position within a long genomic sequence. Unlike prior work, which fixes the convolution hyperparameters (e.g., kernel size and dilation rate) for the model architecture before seeing the task and data, we employ a data-driven approach that automatically learns the architecture configuration from the end task data (see later section). This new approach effectively improves downstream task performance.

### 2.3.3 Transformer Encoder

The transformer encoder, denoted as $g$, takes $h^{\text{DNA}}$ as input and outputs intermediate representations (last hidden states) $h^{\text{intermediate}} = R^{N \times D}$. While L2G is compatible with various language models, we chose RoBERTa-base in our experiments in this work because its model size is smaller than or comparable to most transformer-based genomic FMs, such as DNABERT, DNABERT-2, Enformer, and Nucleotide Transformer-500M. This choice ensures a fair comparison of different methods. The embedding dimension $D$ for RoBERTa-base is 768.

### 2.3.4 Linear Predictor

The predictor, denoted as $p^t$, takes $h^{\text{intermediate}}$ as input and returns a task-specific output tensor. The goal of $p^t$ is to map the learned representations to the desired output dimension. Following ORCA (Shen et al., 2023), we use average pooling along the sequence length dimension. A single linear layer then transforms the pooled outputs of the language models to produce the final prediction.

### 2.4 Model Training

L2G is trained in three stages: neural architecture search, embedder pre-training, and fine-tuning (see **Algorithm** 1 and **Fig.** 2B).

### 2.4.1 Neural Architecture Search

Neural Architectural Search is a machine learning technique that automates the design of deep neural network architectures. Instead of relying on substantial manual efforts by human experts, NAS identifies architectures that perform well

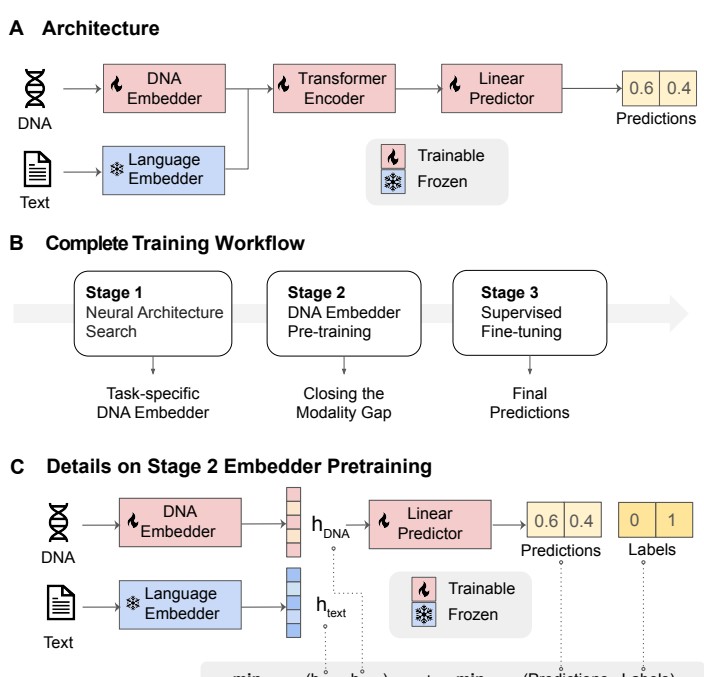

Figure 2: The overall workflow. **A**. The architecture of L2G. The architecture of the L2G model is composed of a CNN embedder, transformer layers from a pre-trained language model, and a linear predictor. **B**. The model is trained in three stages. In stage 1, L2G performs a Neural Architecture Search to optimize the embedder architecture for a given task. In stage 2, the CNN embedder is pre-trained to minimize the modality gap between DNA embeddings and language embeddings. In stage 3, the entire model is fine-tuned on task-specific data in a supervised manner by minimizing the task-specific loss between the final predictions and the true labels. **C**. In stage 2, L2G closes the modality gap by pertaining the embedder with a joint objective. The CNN embedder is attached to a linear predictor and trained with a joint objective, which simultaneously minimizes a) the distribution distance between language embeddings ($h_{\text{text}}$) and DNA embeddings ($h_{\text{DNA}}$), and b) the task-specific loss between predictions from the linear predictor and the true labels. After the embedder pretraining, we retain only the encoder, attach it to the transformer backbone and a new linear predictor, and then perform final fine-tuning.

on a given task through algorithmic solutions (Liu et al., 2018). Given its success in certain genomics applications (Zhang et al., 2021), we utilize NAS to tailor embedder architectures to different tasks.

---

**Algorithm 1** Pseudocode for the L2G workflow.

---

**Input:** Genomic Dataset $G$, Set of Embedder Backbone Architectures $B$, Language Model $L$, Alignment Loss Weight $\alpha$, Task-Specific Loss Weight $\beta$

**for** *each architecture* $b \in B$ **do**
    Initialize $b$
    $val\_score_b \leftarrow$ Train $b$ for one epoch on $G$

$best\_b \leftarrow \arg\max_{b \in B} val\_score_b$ ;       `// Select the embedder backbone with the best validation score`
$(k, d) \leftarrow \mathrm{DASH}(best\_b)$ ; `// Optimize the kernels and dilations`
$h\_text \leftarrow$ Inference L on the source text dataset ;     `// Generate text embeddings`
Initialize $best\_b$ with $(k, d)$
**for** *epoch* $\in$ *embedder_epochs* **do**
    $pred\_1, h\_DNA \leftarrow best\_b(G)$
    $loss\_1 \leftarrow \mathcal{L}_{MMD}(h\_text, h\_DNA)$
    $loss\_2 \leftarrow \mathcal{L}_{task}(pred\_1, labels)$
    embedder $\leftarrow \min(\alpha \cdot loss\_1 + \beta \cdot loss\_2)$
model $\leftarrow$ embedder + transformer blocks from $L$ + linear predictor
$pred\_2 \leftarrow$ Train model on $G$
**return** $pred\_2$

---

To achieve optimal performance across various downstream tasks, we use a two-step process for selecting the embedder network in L2G. First, we select an optimal *backbone* CNN architecture from a pre-defined search space. In this study, we consider ResNet (He et al., 2016) and UNet (Ronneberger et al., 2015), both of which have been effectively applied in genomics. ResNet, with its deep residual connections, is well-suited for classification tasks that require capturing hierarchical features from sequential data. In contrast, UNet's U-shaped encoder-decoder structure with skip connections makes it particularly effective for dense prediction tasks. Each architecture is trained for one epoch, and the one achieving the highest validation score is selected as the backbone.

Next, L2G applies NAS to optimize *layer operations* for the specific task. Specifically, we learn the optimal kernel size and dilation rate for each convolutional layer in the CNN using the DASH (Diverse-task Architecture Search) algorithm (Shen et al., 2022), which has demonstrated state-of-the-art performance among AutoML methods on the DeepSEA dataset. After this two-step process, both the backbone architecture and the convolutional layers of the embedder $f^t$ are tailored for the target task, effectively capturing meaningful target embeddings from genomics datasets.

### 2.4.2 Embedder Pre-training

The embedder pre-training stage is critical for minimizing the modality gap between DNA and the pre-trained language models, enabling cross-modal adaptation. We propose a joint objective to address the training limitations discussed earlier. The first objective minimizes the distribution distance between DNA and text data, performing modality alignment. Unlike ORCA (Shen et al., 2023), which uses OTDD loss, we utilize Maximum Mean Discrepancy (MMD) as the distance metric in L2G due to its better empirical performance in our ablation studies (**Table 9**).

Additionally, we introduce a second objective – a task-specific loss ($\mathcal{L}_{\text{task}}$) – during embedder pre-training. This enables the embedder to incorporate class information while performing distribution alignment. This task-specific loss is either cross-entropy loss for classication tasks or Mean Squared Error (MSE) for regression tasks. By including $\mathcal{L}_{\text{task}}$, the embedder learns to model the class information effectively when mapping genomics data to the language model's embedding space. Training with only alignment loss but not task-specific loss can result in worse downstream performance, as shown in the ablation studies in the **Results** section.

In summary, the embedder pre-training objective is defined as:

$$\mathcal{L}_{\text{total}} = \alpha \mathcal{L}_{\text{MMD}}(h_{\text{language}}, h_{\text{DNA}}) + \beta \mathcal{L}_{\text{task}}(Y, \hat{Y}),$$

where $\alpha$ and $\beta$ are weights for the MMD loss and task-specific loss, respectively. To optimize performance, we set up a scheduler for these weights, minimizing the task-specific loss first before minimizing the joint objective. We did this because empirically, introducing the alignment loss in later epochs of pre-training embedders achieves the best performance. Specifically, we set the alignment loss weight $\alpha = 0$ during the first two thirds of the total epochs of pre-training, effectively optimizing only the task-specific loss $\mathcal{L}_{\text{task}}$ in the early stage. We then set $\alpha$ to a positive value and optimize the full joint objective. This staged approach empirically improves performance, as delaying the alignment loss allows the embedder to first learn meaningful task-relevant representations before enforcing modality alignment via $\mathcal{L}_{\text{MMD}}$.

### 2.4.3 Fine-tuning

After pre-training the embedder, the entire model – including embedder, transformer encoder, and linear predictor – is fine-tuned using task-specific loss on the target data. To optimize the hyperparameter configuration (e.g., learning rate, dropout rate, weight decay) for fine-tuning, we use ASHA (Li et al., 2020).

## 3 Results

### 3.1 Overview

Directly applying transformer models trained on natural language data to out-of-domain tasks like genomics can degrade the quality of the pre-trained weights, resulting in inefficiencies and inaccuracies due to the fundamental mismatch between the two modalities. To address this, we developed L2G, an effective and efficient workflow designed to repurpose pre-trained language models for genomics tasks through cross-modal transfer learning. Unlike traditional in-modal transfer learning, where transformer models are first pre-trained on large-scale DNA sequencing data before fine-tuning, our approach is not only more competitive in prediction quality but also significantly more efficient. L2G eliminates the need for large-scale self-supervised pre-training, reducing both data and computational requirements while still generalizing effectively across a variety of genomics tasks through fine-tuning.

We demonstrate the empirical effectiveness and efficiency of L2G through extensive experiments on two genomics benchmarks and a challenging regression task for enhancer activity prediction. Beyond presenting results on predictive accuracy, we assess L2G's ability to learn relevant TF motifs and evaluate the efficacy of cross-modal fine-tuning through embedding analyses and ablation studies.

### 3.2 L2G matches or outperforms fine-tuned genomic FMs

Predicting the regulatory function of non-coding DNA based on its sequence is crucial for prioritizing functional non-coding variants and remains a major challenge in genomics (Hill et al., 2023; Zhou & Troyanskaya, 2015). We evaluated L2G on two existing benchmarks, Genomic Benchmarks (Grešová et al., 2023) and the Nucleotide Transformer Benchmarks (Dalla-Torre et al., 2023), to demonstrate its generalizability and efficacy.

We first evaluated L2G on the Nucleotide Transformer Benchmarks (Dalla-Torre et al., 2023), one of the most widely used benchmarks for genomic FMs. This benchmark suite includes eighteen tasks for predicting regulatory elements from four categories: enhancers, promoters, epigenetic marks, and splice sites from DNA sequences with lengths ranging from 300 to 600 bp (**Table** 3). We compared L2G against several representative genomic FMs, including Enformer (Avsec et al., 2021a), DNABERT-1 (Ji et al., 2021), DNABERT-2 (Zhou et al., 2023b), HyenaDNA (1kb) (Nguyen et al., 2024), Nucleotide Transformer - Multispecies (2.5B) and Caduceus-ph (Schiff et al., 2024), which outperforms Caduceus-ps in most of the tasks (Dalla-Torre et al., 2023). All these models have been pre-trained and then fine-tuned.

The complete benchmarking results are displayed in **Table** 4 and **Fig.** 3. L2G achieves the best results on ten tasks and ranks second

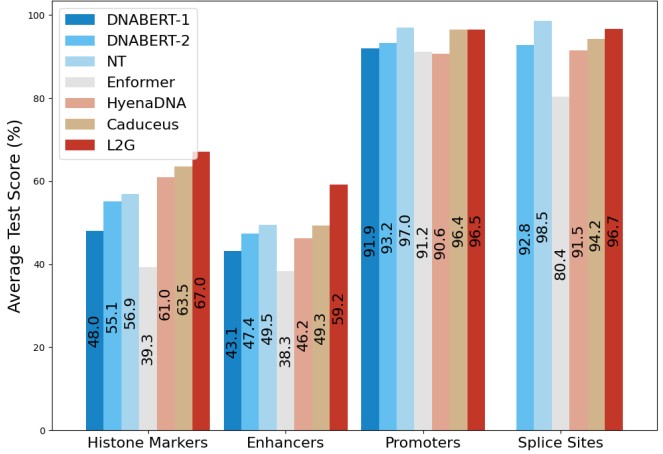

Figure 3: Average performance across task types (histone marks, enhancers, promoters, and splice sites) on the Nucleotide Transformer benchmarks. Each bar represents the average test score of a model. L2G outperforms all other models on histone marks and enhancers tasks and ranks second on promoters and splice sites prediction tasks.

on six others. It demonstrates a clear advantage
in predicting histone marks and enhancers from DNA sequences, outperforming all other genomic FMs. For
promoter and splice site prediction tasks, the Nucleotide Transformer is the top-performing model, followed
by L2G. It is worth noting that the Nucleotide Transformer is the largest model evaluated on this bench-
mark, with 2.5 billion parameters and the most extensive pre-training data, while L2G uses significantly
fewer parameters and no pre-training data. Overall, L2G achieves the highest average test score on the
Nucleotide Transformer benchmark with the least training (pre-training + fine-tuning) data, underscoring
its ability to generalize effectively across tasks even with limited data.

The Genomic Benchmarks dataset (Grešová et al., 2023) includes eight classification tasks: seven binary and
one three-way classification task. These tasks focus on predicting regulatory elements such as promoters,
enhancers, and open chromatin regions from several species, including humans and *Drosophila* (Grešová
et al., 2023; Marin et al., 2023; Nguyen et al., 2024). The inputs sequences have median lengths ranging
from 200 to 2,381 bp. This benchmark includes three baselines: a supervised CNN model, a supervised
transformer model, and a fine-tuned genomic FM, HyenaDNA (Nguyen et al., 2024). The results are shown
in **Table** 5. L2G outperforms all other models in five out of eight tasks and is the second best in the
remaining three, slightly behind HyenaDNA. As demonstrated by the aggregated results using performance
profiles **Fig.** 6C, L2G achieves the best overall performance on the Genomic Benchmarks. Additionally, we
evaluated L2G on a challenging cell-type-specific element classification task from the very recent DART-
Eval (Patel et al., 2024) benchmark, with the results shown in **Table** 6. L2G outperformed all fine-tuned
genomic FM baselines and the *ab initio* baselines.

Overall, L2G matches or outperforms fine-tuned genomic FMs across a variety of regulatory element predic-
tion tasks. This is particularly significant as L2G does not rely on extensive pre-training on unsupervised
DNA data, a standard practice for most genomic FMs. These results highlight the efficacy of the cross-
modal transfer learning approach employed by L2G, which effectively leverages the pre-trained knowledge
embedded in language models to address genomics tasks.

### 3.3  L2G reveals transcription factor motifs

While we have demonstrated the strong benchmark performance of L2G, we also sought to showcase its
utility in downstream applications, such as discovering functional regulatory syntax. Here, we focused on
a regression task to predict the activities of developmental and housekeeping enhancers (de Almeida et al.,
2022) from DNA sequences. Using the DeepSTARR dataset (de Almeida et al., 2022), which predicts
enhancer activity for two distinct promoters in *Drosophila* S2 cells, we compared L2G to baseline methods.

The comparison included the DeepSTARR model, an adaptation of the Basset convolutional neural net-
work (Kelley et al., 2016), as well as several genomic FMs. Full results are presented in **Table** 7. **Fig.** 4A
shows that predictions by L2G align well with measured values for both developmental (PCC=0.66) and
housekeeping (PCC=0.76) enhancers. Compared to other models, L2G outperforms all others in the house-
keeping enhancer prediction task and ranks second in the developmental enhancer prediction task, slightly
behind DeepSTARR (PCC=0.68).

To determine whether L2G learned regulatory syntax, we quantified how each nucleotide contributes to
predicted enhancer activities using DeepLiftShap (Scott et al., 2017) and identified predictive sequence
patterns with TF-Modisco-lite (Shrikumar et al., 2018). Full motif results are shown in **Fig.** 8 and **Fig.** 9.
Interestingly, although both L2G and DeepSTARR achieve high PCC values in predicting developmental
and housekeeping enhancer activities, they identified different sets of TF motifs (**Fig.** 4B).

For developmental TF motifs, both models identified AP-1, GATA and SREBP, but L2G uniquely revealed
the da motif. The daughterless (da) gene, part of the basic helix-loop-helix (bHLH) family, is essential for
several developmental pathways, including sex determination and neurogenesis (Caudy et al., 1988). For
housekeeping TF motifs, L2G uniquely identified BEAF-32 and CRP. The *Drosophila* Boundary Element-
Associated Factor (BEAF) of 32kDa primarily binds near the promoters of numerous housekeeping genes,
contributing to chromatin domain boundary activity and promoter function (Jiang et al., 2009; Bushey et al.,
2009). Similarly, the CRP motif may be associated with housekeeping promoters (Zhimulev et al., 2024).

We acknowledge that differences in identified motifs may arise from several factors. First, we used different implementations of DeepLIFTShap and TF-MoDISco algorithms, as DeepSTARR is based on Keras and TensorFlow, while our implementation uses PyTorch. These framework differences can contribute to variations in motif interpretation and sensitivity. Additionally, DeepSTARR's simpler CNN-based architecture is inherently easier to interpret than L2G's CNN-transformer hybrid model, and hyperparameter variations, such as the similarity thresholds for merging patterns, could also affect motif detection.

Nevertheless, we demonstrate that L2G can predict enhancer activities and reveal relevant TF motifs associated with developmental and housekeeping enhancers, suggesting it is effective in identifying important sequence patterns for prediction tasks.

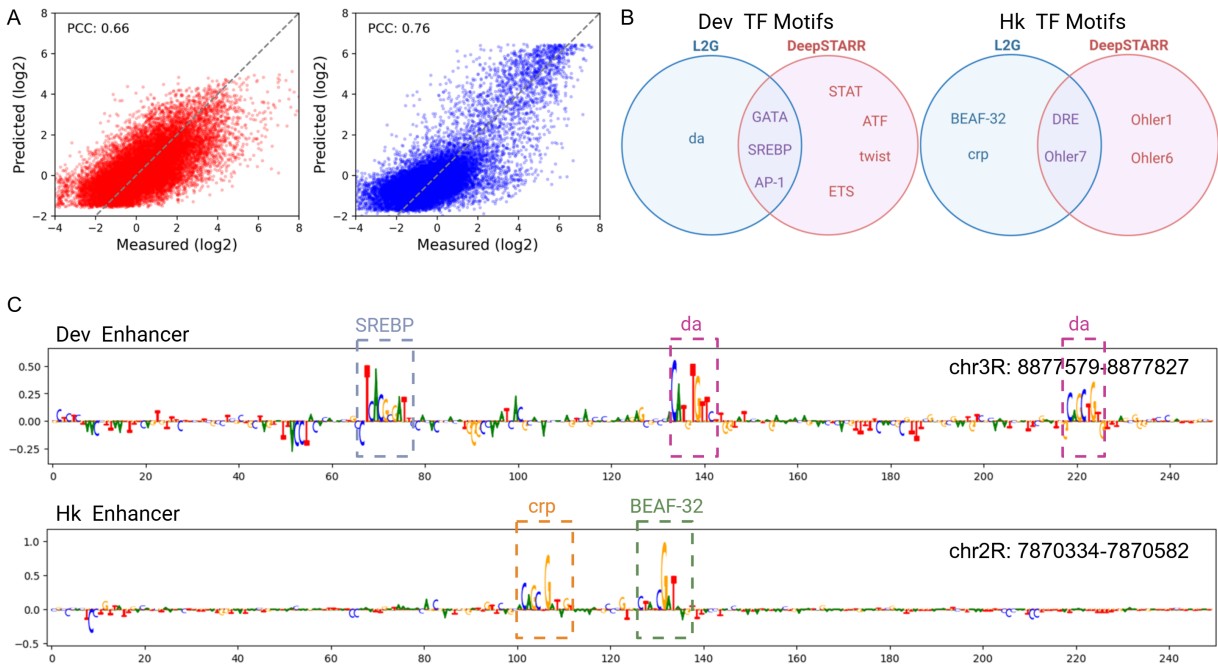

Figure 4: L2G predicts enhancer activities and reveals TF motifs. **A**. Performance of L2G on predicting developmental and housekeeping enhancer activity from DNA sequences in *Drosophila* S2 cells, measured using Pearson Correlation Coefficient (PCC). Scatter plots show predicted vs. observed enhancer activity for developmental (left) and housekeeping (right) enhancers. **B**. Venn diagrams indicating the common and unique TF motifs identified by L2G (blue) and DeepSTARR (red) for developmental (left) and housekeeping enhancers (right). Motif identified by DeepSTARR were retrieved from (de Almeida et al., 2022), with unknown and redundant motifs excluded. While both methods achieve high PCC values in predicting enhancer activities, they identify different sets of motifs. **C**. Nucleotide contribution scores for strong developmental (top) and housekeeping (bottom) enhancer sequences, respectively.

### 3.4   L2G closes the modality gap

L2G bridges the modality gap through joint loss optimization during the embedder pre-training step, simultaneously aligning distributions between text and DNA while optimizing downstream task performance. This approach addresses the challenge where directly fine-tuning language models on genomic tasks often results in weight shifts and poor performance. To evaluate the effectiveness of this strategy, we examined the learned representations of the target modality data generated by different fine-tuning approaches. Specifically, we selected three binary classification tasks from the Nucleotide Transformer benchmark and visualized the learned embeddings for the two classes. We also calculated the Silhouette Score, which quantifies cluster

separation. Scores range from -1 to +1, with +1 indicating well-separated clusters, 0 suggesting overlapping clusters, and -1 indicating incorrect class assignments (Rousseeuw, 1987). Across all three tasks, L2G achieved higher positive Silhouette Scores (**Fig.** 7), demonstrating improved class separation compared to vanilla fine-tuning. This clear distinction in embedding space resulted in better performance.

To better understand the factors contributing to the success of cross-modal fine-tuning in L2G, we conducted a series of ablation studies on selected tasks from the Nucleotide Transformer benchmark, H3, Enhancers, and Promoters TATA.

First, to validate the value of a pre-trained LLM backbone and the effectiveness of cross-modal adaptation in L2G, we compared two configurations: using a pre-trained RoBERTa-base transformer versus a randomly initialized backbone trained from scratch. As shown in **Table** 8 and **Fig.** 5A, L2G with the pre-trained transformer consistently outperforms training from scratch. This confirms that L2G benefits from cross-modal adaptation and that its gains are not simply due to the large size and capacity of the transformer.

We then examined the impact of different distribution alignment metrics, specifically Maximum Mean Discrepancy (MMD) and Optimal Transport Distance for Distributions (OTDD). In principle, OTDD's joint matching of features and labels should capture richer cross-modal structure. However, on our genomics tasks, MMD alignment consistently outperforms OTDD (see **Table** 9 and **Fig.** 5B). We believe this gap stems mainly from OTDD's optimization difficulty: the MMD loss decreases smoothly and converges quickly, whereas OTDD is much harder to learn and yields noisier gradients empirically. Thus, although OTDD offers theoretical appeal for joint-distribution matching, its practical use in genomics-language embedding spaces is hampered by training instability. In contrast, MMD's simpler optimization landscape makes it a more reliable alignment loss in our setting. Addressing OTDD's optimization challenges would be an interesting direction for future work.

Next, we examined the impact of loss functions during embedder pre-training by comparing joint optimization of task-specific and MMD losses against using only the task-specific loss or only the MMD loss. Table 10 and Fig. 5D confirm that joint optimization consistently produces the best performance, demonstrating the benefits of optimizing both alignment and task objectives together during embedder pre-training.

Finally, we evaluated the optimal choice of embedder architecture by comparing the neural architecture search method DASH, used in L2G, with four alternatives: unsearched UNet, unsearched ResNet, the domain-specific CNN model DeepSEA and the embedder from the original ORCA model (Shen et al., 2023). **Table** 11 and **Fig.** 5C indicate that using unsearched UNet or ResNet as embedders already yields strong results, but DASH provides more robust performance and additional gains, although the magnitude of this benefit varies by dataset.

The results of these ablation studies showed that the pre-trained transformer, using MMD as alignment loss, joint loss optimization, and the DASH-based embedder consistently outperformed their respective alternatives (**Fig.** 5). These findings highlight the critical role of pre-training, optimized loss functions, and robust embedder architecture in the success of L2G.

# 4  Discussion

In this work, we investigate the efficacy of cross-modal transfer in genomics. By analyzing a general-purpose cross-modal fine-tuning method, we identified key limitations in both architectural design and objective function. To address these challenges, we introduced L2G, a new method that incorporates a carefully designed architecture and improved alignment between different modalities. This enables L2G to harness the capabilities of pre-trained language models for genomics tasks. Our evaluations across multiple genomics tasks demonstrate superior average performance compared to fine-tuned genomic FMs and domain-specific expert models, notably without requiring large-scale pre-training.

The success of our cross-modal methods raises important questions: Is the current pre-training approach in genomics the most effective? Do we truly need vast amounts of unsupervised genomics data for pre-training? By leveraging pre-trained language models, L2G bypasses the need for extensive unsupervised pre-training, reducing computational and data demands while still achieving competitive performance with

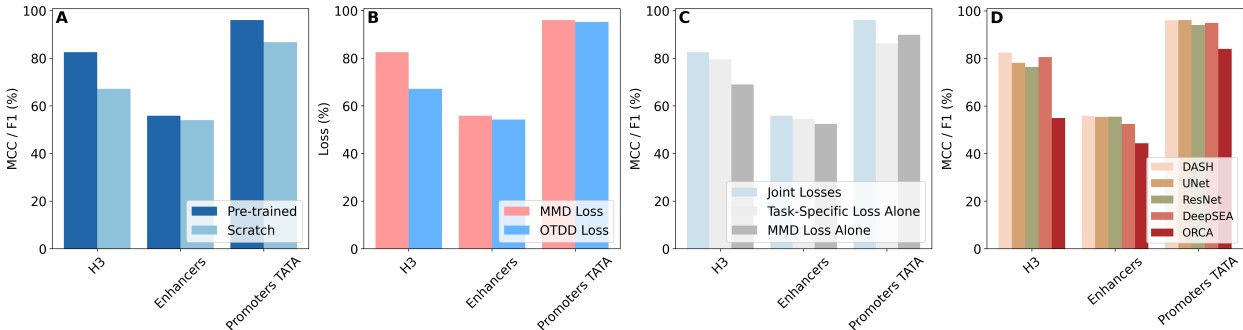

Figure 5: Ablation results for three selected tasks in the Nucleotide Transformer benchmark. **A**: Performance comparison between a pre-trained RoBERTa-base transformer and a randomly initialized backbone trained from scratch. **B**: Evaluation of alignment losses, comparing MMD and OTDD. **C**: Impact of different loss functions during embedder pre-training, comparing joint loss optimization, task-specific loss alone, and MMD loss alone. **D**: Comparison of embedder architectures, including DASH, UNet, ResNet, DeepSEA, and the original ORCA embedder.

in-modal transfer. This challenges the conventional approach of building domain-specific FMs from scratch, suggesting that language models originally developed for NLP can be repurposed for seemingly unrelated domains like genomics. This opens new avenues for cross-disciplinary applications of LLMs, highlighting their versatility and questioning the necessity of developing entirely new models for every domain.

A few contemporaneous studies have also raised concerns about the effectiveness of current genomic FMs. For example, one study observed that genomic FMs offer little to no advantage over traditional models based on one-hot encoded sequences (Tang et al., 2024). Another work found that a supervised-only pipeline named DASHA surpassed the latest genomic FMs on the Nucleotide Transformer benchmark (Xu et al., 2024). Notably, L2G outperforms DASHA on 11/18 tasks in the benchmark and reaches a better average score. At the time of this writing, L2G is the only fine-tuning based approach that outperforms such strong supervised baselines on the Nucleotide Transformer benchmark, despite not being pre-trained on genomics data. Our work along with these contemporaneous studies collectively challenge the prevailing pre-training-then-fine-tuning paradigm for genomic FMs and highlight the need to rethink their development and applications.

Our study has several limitations. First, our evaluations did not cover a broad range of genomics tasks, including long-range prediction tasks that involve more complex interactions and regulatory mechanisms. Future work could extend L2G's application to more complex genomic tasks, such as sequence-based gene expression prediction (Kelley et al., 2018; Avsec et al., 2021a). Second, we have not explored whether the scaling laws common in NLP apply to cross-modal transfer learning for genomics. It remains to be seen whether using increasingly larger language models (trained on natural language) would yield proportionally better performance. Third, interpretability remains a challenge (Chen et al., 2024). While we have used ablation studies and embedding analyses to explain L2G's effectiveness, the underlying mechanisms and interpretability of L2G require further investigation. Lastly, our current cross-modal transfer approach relies on fine-tuning. Future work could explore combining cross-modal transfer with continued pre-training (Gururangan et al., 2020), leveraging both unsupervised text and genomic data to potentially further enhance the performance of domain-specific FMs, albeit at the cost of increased data and computational requirements.

In summary, L2G demonstrates the potential of cross-modal transfer learning to address genomics tasks effectively and efficiently, providing a compelling case for leveraging existing pre-trained models from natural language rather than building domain-specific ones from scratch. This work lays the foundation for further advancements in cross-disciplinary applications of pre-trained language models, extending their utility to a diverse range of biological problems.

## Acknowledgments

This work was supported in part by the National Institutes of Health Common Fund 4D Nucleome Program grant UM1HG011593 (J.M.), National Institutes of Health Common Fund Cellular Senescence Network Program grant UH3CA268202 (J.M.), National Institutes of Health grants R01HG007352 (J.M.), R01HG012303 (J.M.), R21DA061481 (J.M.), and U24HG012070 (J.M.), as well as National Science Foundation grants IIS1705121 (A.T.), IIS1838017 (A.T.), IIS2046613 (A.T.), and IIS2112471 (A.T.). J.M. was additionally supported by the Ray and Stephanie Lane Professorship, a Guggenheim Fellowship from the John Simon Guggenheim Memorial Foundation, a Google Research Award, and a Single-Cell Biology Data Insights award from the Chan Zuckerberg Initiative. A.T. received additional funding from Meta, Morgan Stanley, Amazon, Google, and Scribe. Any opinions, findings, conclusions, or recommendations expressed in this material are those of the author(s) and do not necessarily reflect the views of any of these funding agencies.

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

# A Appendix

## A.1 Code Availability

The source code of L2G can be accessed at: https://github.com/wenduocheng/L2G.

## A.2 Data Availability

In this work, we utilized several public datasets.

- The Genomic Benchmark is available at:
  https://github.com/ML-Bioinfo-CEITEC/genomic_benchmarks.

- The Nucleotide Transformer benchmarks can be downloaded from HuggingFace at:
  https://huggingface.co/datasets/InstaDeepAI/nucleotide_transformer_downstream_tasks.

- The DART-Eval benchmars is available at:
  https://github.com/kundajelab/DART-Eval.

- The DeepSTARR dataset is available on Zenodo at:
  https://doi.org/10.5281/zenodo.5502060.

## A.3 Pre-training resources of various genomic FMs

**Table** 1 provides a comparative summary of the computational resources, model parameters, and pre-training data with various genomic FMs.

| Model | Params | GPUs | Wall clock | Pre-training Data |
|---|---|---|---|---|
| DNABERT (Ji et al., 2021) | 110M | 8-GTX 2080ti-11GB | 25 days | 3.2B |
| DNABERT-2 (Zhou et al., 2023b) | 117M | 8-GTX 2080ti-11GB | 14 days | 32.5B |
| Enformer (Avsec et al., 2021a) | 252M | 64-TPU v3 cores-32TB | 3 days | 14.1B |
| Nucleotide Transformer (Dalla-Torre et al., 2023) | 2.5B | 128-A100-80GB | 28 days | 174B |
| HyenaDNA (Nguyen et al., 2024) | 32K | 1-A100-40GB | 80 mins | 3.2B |
| Caduceus (Schiff et al., 2024) | 1.9M | ? | ? | 35B |

Table 1: Pre-training resources and data of various DNA foundation models. The pre-training data is reported in nucleotides. The computing resources required for pre-training Caduceus are unknown. Note that Enformer is a supervised model trained for the gene expression prediction task and is not pre-trained on unsupervised genomic sequencing data. However, we included it here as it was used as a DNA FM baseline in the Nucleotide Transformer benchmark.

## A.4 Description of the downstream tasks

Genomic Benchmarks dataset consists of eight classification tasks: seven binary and one three-way, focusing on regulatory elements such as promoters, enhancers, and open chromatin regions from several species, including humans, mouse (*Mus musculus*), and roundworm (*C. elegans*) (Grešová et al., 2023). A three-layer CNN serves as the baseline model in this benchmark, and the HyneaDNA study (Nguyen et al., 2024) included a supervised trained transformer baseline. Inputs are DNA sequences with lengths between 200 to 500 bp, except for the Mouse Enhancer Ensembl dataset, which has the longest inputs (median 2,381 bp; maximum 4,707 bp). Metadata for the tasks included in the Genomic Benchmarks (Grešová et al., 2023) is provided in **Table 2**.

Nucleotide Transformer Benchmarks dataset is another widely used benchmark for evaluating genomic FMs. This benchmark suite, introduced alongside the Nucleotide Transformer (Dalla-Torre et al., 2023), evaluates genomic FMs on 18 classification tasks such as predicting regulatory elements for enhancers (human), promoters (human/mouse), epigenetic marks (yeast), and splice sites (human/multispecies) from DNA sequences 300-600 bp long. Performance metrics for several models – including Enformer (Avsec

| Dataset | Samples | Classes | Max Length | Metric |
|---|---|---|---|---|
| dummy_mouse_enhancers_ensembl | 1,210 | 2 | 4,707 | Accuracy |
| demo_coding_vs_intergenomic_seqs | 100,000 | 2 | 200 | Accuracy |
| demo_human_or_worm | 100,000 | 2 | 200 | Accuracy |
| human_enhancers_cohn | 27,791 | 2 | 500 | Accuracy |
| human_enhancers_ensembl | 154,842 | 2 | 573 | Accuracy |
| human_ensembl_regulatory | 289,061 | 3 | 802 | Accuracy |
| human_nontata_promoters | 36,131 | 2 | 251 | Accuracy |
| human_ocr_ensembl | 174,756 | 2 | 593 | Accuracy |

Table 2: Description of datasets in Genomic Benchmarks. Each dataset is described by the name, the total number of samples, the number of target classes, the maximum sequence length in nucleotides, and the evaluation metric used.

et al., 2021a), DNABERT-1 (Ji et al., 2021), DNABERT-2 (Zhou et al., 2023b), HyenaDNA (Poli et al., 2023), and Nucleotide Transformer (Dalla-Torre et al., 2023) – are included, along side results the recent Caduceus-Ph (Schiff et al., 2024). Metadata for the tasks included in the Nucleotide Transformer Benchmarks (Dalla-Torre et al., 2023) is presented in **Table 3**.

| Dataset | Samples | Classes | Max Length | Metric |
|---|---|---|---|---|
| H3 | 13,468 | 2 | 500 | MCC |
| H3K4me1 | 28,509 | 2 | 500 | MCC |
| H3K4me2 | 27,614 | 2 | 500 | MCC |
| H3K4me3 | 33,119 | 2 | 500 | MCC |
| H3K9ac | 25,003 | 2 | 500 | MCC |
| H3K14ac | 29,743 | 2 | 500 | MCC |
| H3K36me3 | 31,392 | 2 | 500 | MCC |
| H3K79me3 | 25,953 | 2 | 500 | MCC |
| H4 | 13,140 | 2 | 500 | MCC |
| H4ac | 30,685 | 2 | 500 | MCC |
| Enhancers | 14,968 | 2 | 200 | MCC |
| Enhancer types | 14,968 | 3 | 200 | MCC |
| Promoter all | 53,276 | 2 | 300 | F1 |
| Promoter TATA | 5,517 | 2 | 300 | F1 |
| Promoter nonTATA | 47,759 | 2 | 300 | F1 |
| Splice sites all | 27,000 | 2 | 400 | Accuracy |
| Splice sites acceptors | 19,961 | 2 | 600 | F1 |
| Splice sites donors | 19,775 | 2 | 600 | F1 |

Table 3: Description of datasets in Nucleotide Transformer Benchmarks. Each dataset is described by the name, the total number of samples, the number of target classes, the maximum sequence length in nucleotides, and the evaluation metric used. Metrics include MCC (Matthews Correlation Coefficient), F1 score, and accuracy, as used in the Nucleotide Transformer study (Dalla-Torre et al., 2023).

DART-Eval is a recent benchmark that curates biologically significant tasks focused on regulatory DNA (Patel et al., 2024). Among these, learning functional regulatory syntax is one of the most biologically relevant downstream applications. DART-Eval includes a task aimed at learning cell-type-specific regulatory syntax by distinguishing uniquely active elements identified from ATAC-seq (Buenrostro et al., 2015) experiments across five cell lines. We evaluated L2G against both fine-tuned and *ab initio* baselines provided by DART-Eval. The fine-tuned baselines include DNABERT-2 (Zhou et al., 2023b), GENA-LM (Fishman et al., 2025), HyenaDNA (Poli et al., 2023), and Nucleotide Transformer (Dalla-Torre et al., 2023). The *ab initio* baseline, ChromBPNet (Pampari et al., 2024), is a strong non-pretrained supervised model that outperformed all other genomic FMs in this task.

Developmental and Housekeeping Enhancer Activity Predictions is a two-class regression task that predicts enhancer activities for housekeeping and developmental enhancers in *Drosophila* S2 cells using 249 bp sequences. The dataset, sourced from the DeepSTARR project (de Almeida et al., 2022), includes a CNN model baseline. The evaluation metric is the Pearson Correlation Coefficient (PCC).

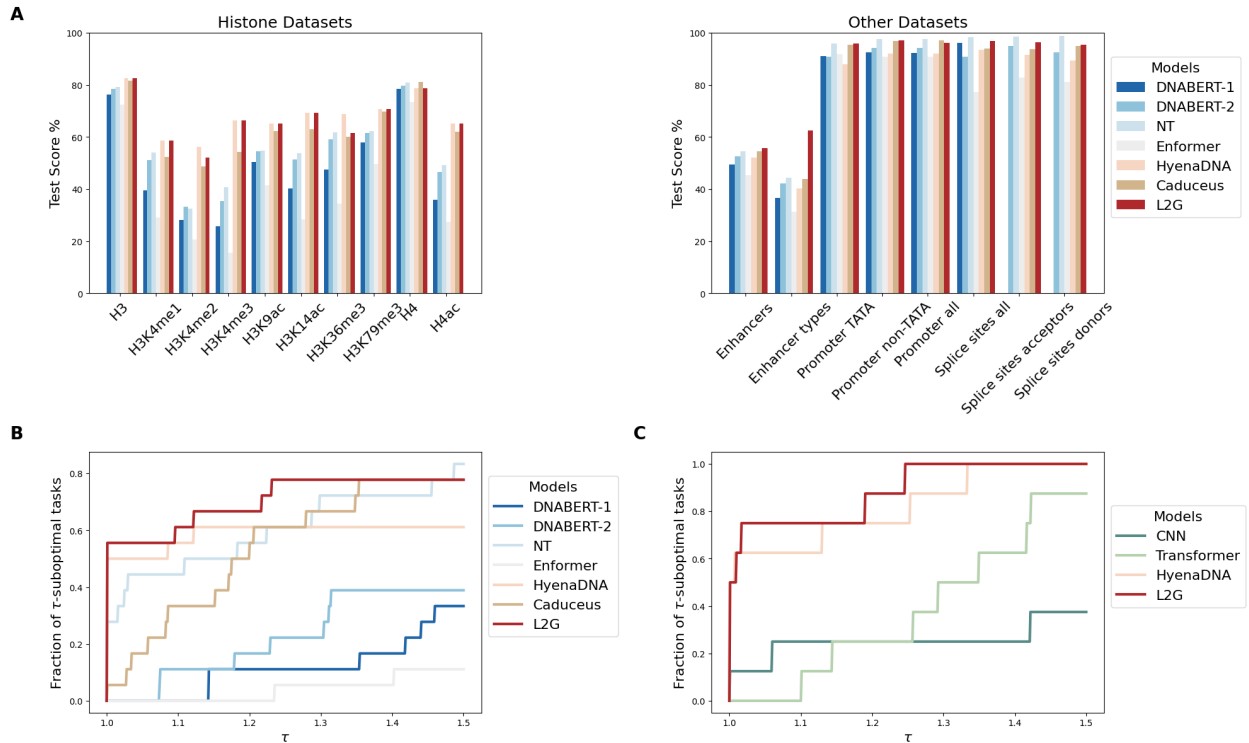

Figure 6: L2G matches or outperforms recent DNA foundation models on the Nucleotide Transformer (NT) Benchmarks and Genomic Benchmarks. **A**. Test scores on the Nucleotide Transformer Benchmarks for histone mark prediction tasks (left) and enhancer, promoter, and splice site prediction tasks (right). The bar for DNA-BERT-1 is missing because it could not predict splice sites. **B**. Aggregating results on the Nucleotide Transformer Benchmarks (**Table** 4) using performance profiles (Dolan & Moré, 2002). Larger values (fractions of tasks on which a method is within a $\tau$-factor of the best) indicate better performance. L2G's curve in the top-left corner demonstrates it is often the best or second best method. **C**. Aggregating results on the Genomic Benchmarks, which include supervised CNN and transformer baselines.

## A.5 Complete results

The complete results on the Nucleotide Transformer Benchmarks are shown in **Table 4** and **Fig. 6A**. We used a batch size of 64 and cross-entropy loss across all datasets. Test scores for other genomic FMs are from Supplementary Table 6 of Dalla-Torre et al.(Dalla-Torre et al., 2023). Results for Caduceus-Ph (Schiff et al., 2024), which outperforms the Caduceus-Ps on 17 out of 18 tasks, are included. To provide a holistic comparison of methods across all datasets, we utilized performance profiles (Dolan & Moré, 2002). Each curve shows the proportion of problems it solves within varying thresholds of a performance factor $\tau$. As shown in **Fig. 6B**, L2G achieves the best or second-best performance across all tasks.

The complete results on the Genomic Benchmarks are shown in **Table 5**. We used a batch size of 64 and cross-entropy loss across all datasets. We trained the CNN and HyenaDNA (32k) baselines, while results for the transformer baseline were obtained from the HyenaDNA paper (Nguyen et al., 2024), as the code is not open-sourced. We also computed the performance profiles for the results on the Genomic Benchmarks. As shown in **Fig. 6C**, L2G achieves top performance across all tasks.

The complete results for the cell-type-specific element classification task on the DART-Eval benchmark are shown in **Table 6**. L2G outperforms both fine-tuned and *ab initio* baselines in overall accuracy. Notably, L2G is the only fine-tuned-based method to surpass the non-pre-trained supervised model ChromBPNet.

The complete results for the Developmental and Housekeeping Enhancer Activity Prediction Task are presented in **Table 7**. For this task, we used a batch size of 128 and mean squared error (MSE) loss. To benchmark the performance of L2G, we included the expert model, DeepSTARR, which was a CNN specifically designed for this task. Additionally, we compared with two genomic FMs, HyenaDNA (32k) and Nucleotide Transformer (v2, 500M). PCC was used as the evaluation metric.

| Dataset | NT | Enformer | DNABERT-1 | DNABERT-2 | HyenaDNA | Caduceus-Ph | L2G |
|---|---|---|---|---|---|---|---|
| *Histone Markers* | | | | | | | |
| H3 | 79.3 | 72.4 | 76.3 | 78.5 | 78.1 | 81.5 | **82.5** |
| H3K4me1 | 54.1 | 29.1 | 39.6 | 51.2 | 51.2 | 52.3 | **58.6** |
| H3K4me2 | 32.4 | 20.7 | 28.2 | 33.3 | 45.5 | 48.7 | **56.2** |
| H3K4me3 | 40.8 | 15.6 | 25.8 | 35.3 | 55.0 | 54.4 | **66.3** |
| H3K9ac | 54.7 | 41.5 | 50.5 | 54.5 | 58.6 | 62.2 | **65.1** |
| H3K14ac | 53.8 | 28.4 | 40.3 | 51.5 | 60.8 | 63.1 | **69.4** |
| H3K36me3 | 61.8 | 34.5 | 47.4 | 59.1 | 61.4 | 60.1 | **68.8** |
| H3K79me3 | 62.3 | 49.8 | 57.8 | 61.5 | 66.9 | 69.7 | **70.7** |
| H4 | 80.8 | 73.5 | 78.4 | 79.7 | 76.3 | **81.1** | 78.8 |
| H4ac | 49.2 | 27.5 | 35.9 | 46.5 | 56.4 | 62.1 | **65.1** |
| **Average** | 56.9 | 39.3 | 48.0 | 55.1 | 61.0 | 63.5 | **67.0** |
| *Enhancer* | | | | | | | |
| Enhancers | 54.5 | 45.4 | 49.5 | 52.5 | 52.0 | 54.6 | **55.8** |
| Enhancer types | 44.4 | 31.2 | 36.7 | 42.3 | 40.3 | 43.9 | **62.6** |
| **Average** | 49.5 | 38.3 | 43.1 | 47.4 | 46.2 | 49.3 | **59.2** |
| *Promoter* | | | | | | | |
| Promoter TATA | 95.9 | 91.8 | 91.0 | 90.9 | 87.9 | 95.3 | **96.0** |
| Promoter non-TATA | **97.7** | 90.9 | 92.4 | 94.3 | 91.9 | 96.9 | 97.2 |
| Promoter all | **97.5** | 90.9 | 92.2 | 94.3 | 91.9 | 97.0 | 96.2 |
| **Average** | **97.0** | 91.2 | 91.9 | 93.2 | 90.6 | 96.4 | 96.5 |
| *Splice Sites* | | | | | | | |
| Splice sites all | **98.2** | 77.2 | 96.2 | 90.9 | 93.4 | 94.0 | 97.9 |
| Splice sites acceptors | **98.6** | 82.9 | / | 94.9 | 91.6 | 93.7 | 96.4 |
| Splice sites donors | **98.7** | 81.2 | / | 92.5 | 89.4 | 94.8 | 95.5 |
| **Average** | **98.5** | 80.4 | / | 92.8 | 91.5 | 94.2 | 96.7 |

Table 4: The performance of each model on the Nucleotide Transformer Benchmarks. Metrics used by task: MCC for histone markers, F1-score for enhancers and splice site acceptors/donors, and accuracy for splice site all. Bold indicates the best performance, and underline indicates the second-best. NT stands for nucleotide transformer (multispecies, 2.5B version). The results for other baselines are retrieved from the Nucleotide Transformer paper (Dalla-Torre et al., 2023). DNABERT-1 could not be trained on two splice site prediction tasks because the input sequence length exceeded the maximum context length allowed by DNABERT-1.

| Dataset | CNN | Transformer | HyenaDNA | L2G |
|---|---|---|---|---|
| Mouse Enhancers | 72.8 | 80.1 | **82.6** | 79.3 |
| Coding vs Intergenomic seqs | 88.2 | 88.8 | 89.6 | **91.7** |
| Human vs Worm | 92.8 | 95.6 | **96.5** | **96.5** |
| Human Enhancers Cohn | 71.6 | 70.5 | 73.0 | **73.2** |
| Human Enhancers Ensembl | 80.2 | 83.5 | 86.9 | **88.4** |
| Human Ensembl Regulatory | 93.9 | 91.5 | 92.0 | **93.9** |
| Human Nontata Promoters | 85.8 | 87.7 | **94.3** | 92.9 |
| Human OCR Ensembl | 67.8 | 73.0 | **79.1** | 78.9 |
| **Average** | 81.7 | 83.8 | 86.8 | **86.9** |

Table 5: The performance of each model on the Genomic Benchmark dataset. The evaluation metric is accuracy (the higher, the better). Bold indicates the best performance, and underline indicates the second-best.

| Model | Overall | GM12878 | H1ESC | HEPG2 | IMR90 | K562 |
|---|---|---|---|---|---|---|
| ChromBPNet-like | 0.667 | **0.887** | 0.886 | 0.874 | 0.874 | 0.865 |
| DNABERT-2 | 0.650 | 0.878 | **0.886** | 0.875 | 0.863 | 0.836 |
| GENA-LM | 0.636 | 0.869 | 0.881 | 0.874 | 0.858 | 0.836 |
| HyenaDNA | 0.610 | 0.866 | 0.868 | 0.862 | 0.853 | 0.838 |
| Nucleotide Transformer | 0.632 | 0.868 | 0.881 | 0.871 | 0.859 | 0.836 |
| L2G | **0.690** | 0.879 | 0.876 | **0.878** | **0.877** | **0.871** |

Table 6: Cell-type-specific element classification task from the DART-Eval benchmark. We report the overall accuracy across all classes and the accuracy specific to each cell line. Bold text indicates the best performance, and underlined text indicates the second-best. L2G is the only fine-tuned method that outperformed the non-pretrained supervised model ChromBPNet.

| Dataset | HyenaDNA | Nucleotide Transformer | DeepSTARR | L2G |
|---|---|---|---|---|
| dev | 0.57 | 0.64 | **0.68** | 0.66 |
| hk | 0.65 | 0.75 | 0.74 | **0.76** |
| Mean | 0.61 | 0.70 | **0.71** | **0.71** |

Table 7: The performance of each model on the *Drosophila* enhancers prediction regarding the developmental (dev) and housekeeping activity (hk).

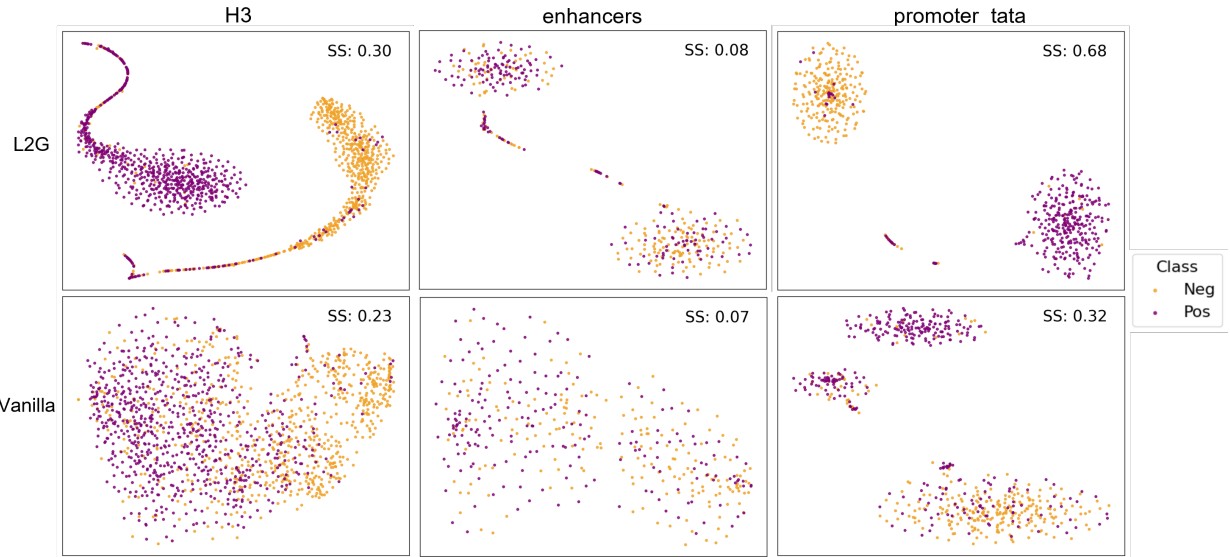

Figure 7: Visualization of the learned embedding of target modality data for models trained with L2G (top), and vanilla fine-tuning (bottom). The target modality data is from three representative downstream tasks from the NT Benchmark: histone modification (H3, left), enhancer regions (enhancers, middle), and promoter regions (promoter_tata, right). Pink dots represent positive class, while yellow dots indicate negative class. Each plot includes the Silhouette Score, a widely used metric for evaluating cluster separation. The score ranges from -1 to +1, where: +1 indicates well-separated clusters, 0 suggests indifferent clusters, and -1 indicates potential misclassification of points between clusters (Rousseeuw, 1987).

## A.6   Embedding analysis

To evaluate the quality of the learned representations of the target modality data, we analyzed the embeddings generated by L2G and compared them to those obtained through vanilla fine-tuning, which refers to directly training the model with task-specific loss without the NAS or embedded pertaining steps. This analysis was conducted on three binary classification tasks from the Nucleotide Transformer benchmarks: H3, enhancers, and promoter_tata. We visualized the embeddings of the two classes in each task using t-SNE, as shown in **Fig. 7**. This visualization provided a qualitative assessment of how well the embeddings separate the classes in a reduced-dimensional space. To quantitatively measure cluster separation, we calculated the Silhouette Score for each set of embeddings. The Silhouette Score ranges from -1 to +1, where +1 indicates well-separated clusters, 0 signifies overlapping clusters, and -1 suggests incorrect class assignments (Rousseeuw, 1987). Across all three datasets, L2G consistently achieved higher Silhouette Scores compared to vanilla fine-tuning, demonstrating its superior ability to produce distinct class separations in the embedding space.

## A.7 Ablation studies

We conducted a series of ablation studies to evaluate the contributions of different components in our method. Specifically, we assessed the impact of pre-trained transformers, the losses during embedder pretraining, and the embedder architecture. The results on three tasks from the Nucleotide Transformer benchmarks are shown. Detailed results are provided in **Table** 8 for ablation study for the pre-training Body, **Table** 9 for ablation study for the distribution alignment metrics, **Table** 10 for ablation study for the losses during embedder pre-training, **Table** 11 for the embedder architecture, **Table** 12 for different training strategies, **Table** 13 for different transformer model backbones, and **Table** 14 for different source language data.

| Dataset | Pre-trained | Scratch |
|---|---|---|
| H3 | 82.5 | 67.1 |
| Enhancers | 55.8 | 54.0 |
| Promoter TATA | 96.0 | 86.7 |

Table 8: Performance comparison of pre-trained RoBERTa with L2G versus randomly initialized RoBERTa trained from scratch on three datasets from the Nucleotide Transformer benchmark.

| Dataset | MMD | OTDD |
|---|---|---|
| H3 | 82.5 | 67.1 |
| Enhancers | 55.8 | 54.7 |
| Promoter TATA | 96.0 | 95.2 |

Table 9: Performance comparison of Maximum mean Discrepancy (MMD) and Optimal Transport Dataset Distance (OTDD) as alignment metric used during the embedder pre-training stage on three selected datasets on the Nucleotide Transformer benchmark.

| Dataset | Joint losses | Task-specific loss only (alpha=0) | MMD loss only (beta=0) |
|---|---|---|---|
| H3 | 82.5 | 79.5 | 69.0 |
| Enhancers | 55.8 | 54.5 | 52.4 |
| Promoter TATA | 96.0 | 86.2 | 89.8 |

Table 10: Performance comparison of different losses used during the embedder pre-training stage on three selected datasets on the Nucleotide Transformer benchmark.

## A.8 Motif analysis

We calculated the nucleotide contribution scores using a DeepLiftShap method from Catum GitHub Repository for developmental and housekeeping enhancer activities. DeepLiftShap combines the DeepLIFT (Shrikumar et al., 2017) algorithm with SHAP (SHapley Additive exPlanations) values to attribute model predictions to input features by calculating the contribution relative to a reference baseline (Scott et al., 2017). Each feature is a nucleotide at a specific position.

Following the DeepSTARR methodology (de Almeida et al., 2022), we used 100 dinucleotide-shuffled versions of each input sequence as baseline sequences. Hypothetical importance scores for each sequence were multiplied by its one-hot encoded matrix to derive the final nucleotide contribution scores.

Motifs were identified using TF-Modisco-lite, a more efficient implementation of TF-Modisco (Shrikumar et al., 2018), on the nucleotide contribution scores for each enhancer type separately (de Almeida et al., 2022). For motif annotation, we downloaded two reference databases for *Drosophila*: OnTheFly (Shazman et al., 2014) and FlyFactorSurvey (Zhu et al., 2011), from the MEME suite (Bailey et al., 2015), and compared motifs using TOMTOM (Gupta et al., 2007). Motifs with fewer than 30 seqlets were discarded. The resulting motifs for developmental and housekeeping enhancers are visualized in **Fig.** 8 and **Fig.** 9, respectively,

| Dataset | DASH | UNet | ResNet | DeepSEA | ORCA |
|---|---|---|---|---|---|
| H3 | 82.5 | 78.1 | 76.4 | 80.5 | 54.9 |
| Enhancers | 55.8 | 55.4 | 55.5 | 52.4 | 44.3 |
| Promoters TATA | 96.0 | 96.1 | 94.0 | 94.9 | 84.0 |

Table 11: Performance comparison of different embedder architectures on three selected datasets from the Nucleotide Transformer benchmark. The architectures compared include DASH, unsearched UNet and ResNet, DeepSEA (a three-layer CNN), and the embedder from the original ORCA model (a single-layer convolutional model). DASH, through neural architecture search, optimizes the configuration of UNet and ResNet backbones, while the unsearched versions of UNet and ResNet use a fixed kernel size of 3 and a fixed dilation rate of 1.

| Dataset | Train | Frozen | No |
|---|---|---|---|
| H3 | 82.5 | 74.9 | 76.0 |
| Promoters TATA | 96.0 | 95.1 | 92.8 |
| Enhancers | 55.8 | 54.9 | 48.0 |

Table 12: Performance comparison of three strategies on three datasets from the Nucleotide Transformer benchmark. To isolate the contribution of the pre-trained transformer trunk, we evaluate: (1) full fine-tuning, where all transformer parameters are trainable; (2) frozen transformer backbone, where all transformer weights are frozen; and (3) no-transformer, where only the encoder and predictor are trained.

| Dataset | RoBERTa-Base | RoBERTa-Large |
|---|---|---|
| H3 | 82.5 | 85.5 |
| Promoters TATA | 96.0 | 96.2 |
| Enhancers | 55.8 | 57.1 |

Table 13: Performance comparison between using pre-trained RoBERTa-Base (149M parameters) and RoBERTa-Large (387M parameters) as transformer backbones on three selected datasets from the Nucleotide Transformer benchmark.

| Dataset | CONLL2003 | JNLPBA |
|---|---|---|
| H3 | 82.5 | 82.2 |
| Promoters TATA | 96.0 | 96.2 |
| Enhancers | 55.8 | 56.5 |

Table 14: Performance comparison of different language source data on three selected datasets from the Nucleotide Transformer benchmark. We evaluated the impact of language source data by comparing the CoNLL-2003 (Sang & De Meulder, 2003) and JNLPBA (Collier et al., 2004) datasets. We chose JNLPBA because it is specifically designed for biomedical named entity recognition, making it more domain-aligned with our genomic sequence analysis tasks than CoNLL-2003.

Figure 8: Motifs discovered for developmental enhancers by L2G. Details include forward and reverse sequences, seqlet count, motif name, Q-Value, and closest database match. The Q-Value is statistical measure that represents the false discovery rate (FDR) for the motif. Lower Q-values indicate more significant results.

Figure 9: Motifs discovered for housekeeping enhancers by L2G. Details include forward and reverse sequences, seqlet count, motif name, Q-Value, and closest database match.

### A.9 Implementation details

**Table** 15 provides the hyperparameter settings used for training L2G.

| Hyperparameter | Value |
|---|---|
| Distribution Alignment Metric | MMD |
| Transformer Backbone | RoBERTa-base |
| Target Sequence Length | 512 |
| Training Epochs | 25 |
| Embedder Pre-training Epochs | 80-100 |
| Warm-up Epochs | 5 |
| Decay Epochs | 25 |
| $\alpha$ (Weight for Alignment Loss) | 1 |
| $\beta$ (Weight for Task Loss) | 1 |
| Dropout | 0.05 |
| Gradient Clipping | [-1, 1] |
| Batch Size | 64-128 |
| Embedder Pre-training Optimizer | SGD |
| Embedder Pre-training Learning Rate | Searched by DASH |
| Fine-tuning Optimizer | Adam |
| Fine-tuning Optimizer Betas | [0.9, 0.98] |
| Fine-tuning Learning Rate | 1e-5 |
| Weight Decay | 1e-5 |
| Scheduler | Step Decay |

Table 15: Hyperparameter settings for training L2G.

For all tested datasets, we applied data during training by randomly shifting input sequence by up to 3 bp and reverse-complementing sequences. During testing, predictions from the forward and reverse complement sequences were averaged. This approach is commonly used in genomics to improve the prediction accuracy of deep learning models (Zhou & Troyanskaya, 2015; Avsec et al., 2021a).

