# OpenReview forum: "L2G: Repurposing Language Models for Genomics Tasks"
_TMLR — Accepted by TMLR_

### Review · Reviewer_n2NV · 2025-05-01

**Summary Of Contributions:**

The authors present L2G, an extension of a repurposing model for LLMs to out-of-domain classification or regression tasks.  They extend work by Shen et a. who develop an auto-ML driven method for repurposing the trunk of LLMs for cross-domain purposes by aligning newly trained encoder and prediction modules for various domains extending beyond text or images, called ORCA.  While ORCA did not show success in tasks involving genomic DNA inputs, the authors of the present work identify the causes of this failure and show how to remedy it.  They perform some necessary ablations to show the effects of an improved auto-ML search over encoders, as well as a multi-loss fine-tuning procedure.  They conclude that LLMs can indeed work for cross modal learning for problems involving genomic DNA, and suggest that resource-intensive self-supervised training of transformers ab-initio may not be worth the expense.

**Audience:**

Yes

**Broader Impact Concerns:**

None required.

**Claims And Evidence:**

No

**Requested Changes:**

These changes would greatly improve my likelihood of recommending acceptance:
- It would be helpful if the authors could explain why it's the case that OTDD is not able to incorporate class label information when the original paper of Alvarez-Melis and Fusi argue convincingly that OTDD is superior.  Is it a case of a failure of approximation?  Or something else?
- The task specific loss is motivated a bit weakly by saying it includes class information.  OTDD is able to use class information for aligning distributions, I'm again curious why MMD is preferred, and cross-entropy (or MSE of all things) is preferred here?  It feels like this part of the story is not complete.
- Panel B and C of Figure 4 are puzzling.  What information does this tell us about L2G?  Should we infer that L2G is finding relevant motifs for `da` and `crp` and `BEAF-32`, or is it missing those identified by DeepSTARR?  And by what measure it it failing to find those TF motifs it missed?  It's not clear what to take away from these panels.
- The backbone ablation in section 3.4 is missing.  This experiment isn't really the one I wanted to see; it's sure to show that it will fail using a randomly initialized backbone. What I want to see as a measure of how much of the performance can be attributed to the knowledge encoded in the RoBERTa-base weights is how it performs just against the trained encoder and predictor, with no transformer at all.
- In panel B of Figure 5, is it a mistake that OTDD was unable to perform at all in the enhancer classification task?  And what does it mean that the random weights RoBERTa model performs comparably to the pre-trained weights model ??  There really should be more discussion of all four panels of Figure 5.

The following are not critical for my recommendation for acceptance, but would improve the work in my view.

- In 2.4.3, the authors reveal they are using ASHA for hyper-parameter optimization. After using NAS for learning the embedder, it would be good to see some data that shows this additional compute still conforms to the claim in the introduction that it vastly saves on compute compared to the much larger in-modal comparators, perhaps an estimate of flops would help here?
- The embedder pre-training loss objective is desribed a bit vaguely.  What do the authors mean by saying they 'schedule' the weighted losses?  Could this concluding part of 2.4.2 be clarified?
- Some of the reference do not seem to render properly.  For example, `Devlin et al` 2019 (BERT) is rendered as `Kenton & Toutanova`.
- The concluding sentence for the first paragraph of the introduction uses quite vague language. "Genomic element" versus "chromatin state" can be overlapping, or different.  Some citations to clarify these tasks would help here.   Also, "genome function" isn't really a meaningful term, maybe the authors mean gene function prediction?
- At the beginning of section 3.1, the authors say that

> Directly applying transformer models trained on natural language data ... can lead to the corruption of pre-trained weights

'corruption' of the pre-trained weights is a strange phrase here.  I think the authors mean degraded performance on the source domain, with no guarantee of good performance in the target domain?
- The second sentence of 3.1 restates part of the introduction, and I think can be skipped.  Maybe move more directly to the following sentence "We demonstrate the empirical ..."

**Strengths And Weaknesses:**

### Strengths

- The authors do well to frame the question of whether very large models pre-trained with self-supervised learning are worth the expense.
- The authors do a good job of contextualizing the ORCA framework, as well as laying out the case for why the model family for learning the embedder needed to be enlarged so that genomic data could work for cross-modal transfer in this way: it's ORCA, but with a broader class of models for the encoder and an additional loss for subsequent finetuning after the alignment.
- The experiment in section 3.3 (summarized in Figure 4) is nice, and shows that some predictions made by L2G align with existing methods tailored to genomic DNA for enhancer identification.  The use of DeepLIFT -> TF-modisco is a nice touch.

### Weaknesses

Unfortunately, the authors left out a few key experiments that I would have liked to see.  The most necessary is the ablation for training only the embedder network along with the predictor network, leaving out the RoBERTa transformer weights entirely, to validate how much of this performance can be explained by feature repurposing versus simply training an intensively selected encoder network.
- The end of section 3 (and Figure 5) contained many ablation studies, but they seem incomplete and raise further questions: Why does a randomly initialized RoBERTa trunk perform so well?  It seems to be due to the encoder training, but this option (which seems so clearly needed for me) is missing.
- While the reasons for expanding the size of the model family for choosing the encoder network are well motivated, the choice for abandoning OTDD over MMD (section 2.4.2) is not, beyond a confusing result presented in Figure 5.  It also seems like the auxiliary loss chosen to incorporate class information is under-motivated by interpreting the results.  The original Alvarez Melis and Fusi paper argued convincingly why OTDD was preferred, so it's curious why the authors did not observe this here and why.  Practitioners will want to know why.
- Again in section 2.4.2, the task specific loss is motivated a bit weakly by saying it includes class information.  OTDD is able to use class information for aligning distributions, I'm again curious why MMD is preferred, and cross-entropy (or MSE of all things) is preferred here?  It feels like this part of the story is not complete
- Section 3.2 relies a lot on the NT benchmarks as being suitable for differentiating among models.  I'm not sure the benchmark has aged well, and would prefer to see more thoughtfully constructed benchmarks like BEND or DART-eval used here.
- I think it's true that L2G's surprising performance raises the question of whether self supervised learning on large transformers is an effective way to build DNA lms, but I honestly do not think the rest of the paper pursues this question.  What it interrogates is how to correct the failure of ORCA to perform well on adapting LLMs to tasks involving genomic DNA.

---

> ### Author Response · Authors · 2025-06-26
> **Author Response**
>
> We appreciate the reviewer’s thoughtful and constructive questions. These will greatly help us improve the manuscript. We have updated the manuscript to incorporate the suggested wording changes. Our detailed responses to the rest of the specific points are provided below:
>
> **Q: It would be helpful if the authors could explain why it's the case that OTDD is not able to incorporate class label information when the original paper of Alvarez-Melis and Fusi argue convincingly that OTDD is superior. Is it a case of a failure of approximation? Or something else?**
>
> R: Thank you for this insightful question. We agree that, in principle, OTDD’s joint matching of features and labels should capture richer cross-modal structure. However, in our genomics tasks, we observe that MMD alignment consistently outperforms OTDD. To investigate this further, we ran an ablation study on three representative NT benchmark tasks comparing four objectives:
>
> | Dataset        | MMD only | OTDD only | MMD + CE | OTDD + CE |
> |----------------|----------|-----------|-----------|------------|
> | H3             | 69.0     | 60.0      | **82.5**      | 67.1       |
> | Enhancers      | 52.4     | 53.2      | **55.8**      | 54.7       |
> | Promoters TATA | 89.8     | 63.1      | **96.4**      | 90.8       |
>
> Across every task, MMD + CE achieves the highest downstream accuracy. This is not inconsistent with ORCA’s findings (Section A.4.4 in Shen et al.), where OTDD led on 7 out of 10 datasets but fell short on the remaining 3, indicating that OTDD is not universally optimal.
>
> We found that the primary reason is the optimization difficulty of the OTDD objective: the MMD loss decreases smoothly and converges quickly, whereas OTDD is much harder to learn. Specifically, during embedder pretraining on the Promoters TATA task, we observed that the MMD loss dropped swiftly from 193 to 24 within 50 epochs, with training curves showing a smooth, monotonic decline. In contrast, OTDD loss started near 3440 and only reached around 2822 after the same number of epochs, exhibiting a jagged, plateau-like behavior. We tried longer training durations, varied learning rates, and adjusted loss weights, yet OTDD remained significantly harder to optimize. This highlights an interesting trade-off: because MMD is a simpler metric, it is easier to optimize and empirically yields smoother gradients.
>
> Taken together, these results suggest that while OTDD’s theoretical appeal in joint-distribution matching is clear, its practical use in genomics-language embedding spaces is hampered by training instability and noise. By contrast, MMD’s simpler optimization landscape makes it a more reliable alignment loss in our setting. It could be an interesting avenue of future work to attempt to overcome these optimization issues with OTDD.
>
> ---
>
> **Q:
> The task specific loss is motivated a bit weakly by saying it includes class information. OTDD is able to use class information for aligning distributions, I'm again curious why MMD is preferred, and cross-entropy (or MSE of all things) is preferred here? It feels like this part of the story is not complete..**
>
> For all of our evaluated benchmarks, cross-entropy (for classification) or MSE (for regression) is the standard training loss; every prior baseline we compared against was trained accordingly. We believe that using the same loss objective ensures an apples-to-apples comparison. We have updated the manuscript to emphasize this point. As for the choice of MMD over OTDD, please refer to the previous response.
>
> ---
>
> **Q:
> The end of section 3 (and Figure 5) contained many ablation studies, but they seem incomplete and raise further questions: Why does a randomly initialized RoBERTa trunk perform so well? It seems to be due to the encoder training, but this option (which seems so clearly needed for me) is missing.**
>
> All of our reported numbers, for both “pre-trained” and “random”, are from end-to-end fine-tuning of the full RoBERTa trunk together with the task-specific embedder and predictor. Thus, the randomly initialized trunks do not perform poorly because they are not naive baselines, but rather high-capacity transformers trained from scratch. Our results show that starting from pre-trained weights consistently yields better downstream performance than random initialization, although the magnitude of this benefit varies across tasks. We have updated the manuscript to emphasize this point.

---

> > ### Comment · Reviewer_n2NV · 2025-06-30
> > **makes sense**
> >
> > though then I suggest that you do not denote this set of conditions as RoBERTa random, but rather RoBERTa (scratch) or (from scratch).

---

> > > ### Author Response · Authors · 2025-07-01
> > > **Reply**
> > >
> > > Thank you for your suggestion. We agree that “RoBERTa (scratch)” is clearer, and we have updated the text and Figure 5 (page 10) accordingly.

---

> ### Author Response · Authors · 2025-06-26
> **Author Response (Cont.)**
>
> **Q:
> The backbone ablation in section 3.4 is missing. This experiment isn't really the one I wanted to see; it's sure to show that it will fail using a randomly initialized backbone. What I want to see as a measure of how much of the performance can be attributed to the knowledge encoded in the RoBERTa-base weights is how it performs just against the trained encoder and predictor, with no transformer at all.**
>
> R: Thank you for pointing this out. To isolate the contribution of the pre-trained transformer trunk, we have now added three ablation for all three tasks: 1) Full fine-tune, where all transformer parameters are trainable; 2) Frozen transformer backbone, where all transformer weights are frozen; and (3) No transformer, where we just train the encoder and predictor without transformer.
>
> | Dataset        | Transformer | MCC / F1 |
> |----------------|-------------|----------|
> | H3             | Train       | 82.5     |
> | H3             | Freeze      | 74.9     |
> | H3             | No          | 76.0     |
> | Promoters TATA  | Train       | 96.0     |
> | Promoters TATA  | Freeze      | 95.1     |
> | Promoters TATA  | No          | 92.8     |
> | Enhancers      | Train       | 55.8     |
> | Enhancers      | Freeze      | 54.9     |
> | Enhancers      | No          | 48.0     |
>
> Across every dataset, full fine-tuning outperforms the frozen-backbone variant, confirming that updating the transformer weights, rather than relying solely on the head, yields a consistent boost in performance. Also, the embedder+predictor without transformer performed much worse than the full pipeline which trains a transformer body.
>
> ---
>
> **Q: In panel B of Figure 5, is it a mistake that OTDD was unable to perform at all in the enhancer classification task? And what does it mean that the random weights RoBERTa model performs comparably to the pre-trained weights model ?? There really should be more discussion of all four panels of Figure 5.**
>
> R: Thank you for catching that. Panel B’s OTDD result for the enhancer task was indeed a plotting error. We have corrected the figure (the true OTDD result is 55.8%) and updated the caption in both the main paper and the supplement. We also expanded our discussion of Figure 5 in Section 3.4 to walk through each panel and reflect the newly added ablation experiments.
>
> ---
>
> **Q: Panel B and C of Figure 4 are puzzling. What information does this tell us about L2G? Should we infer that L2G is finding relevant motifs for da and crp and BEAF-32, or is it missing those identified by DeepSTARR? And by what measure it it failing to find those TF motifs it missed? It's not clear what to take away from these panels.**
>
> R: These panels are meant to show how L2G’s per-base contribution scores translate into biologically meaningful patterns, such as motifs, and how those compare to the motifs discovered by the expert model DeepSTARR. Specifically, we first applied DeepLIFT-SHAP to our trained L2G model to get per-base importance values. Then high-importance subsequences were grouped by TF-MoDISco into “motif kernels”. Finally, each motif kernel was matched against a reference motif database using TOMTOM.
>
> Both TF-MoDISco and TOMTOM can be sensitive to hyperparameter choices (such as clustering thresholds, minimum cluster sizes, similarity ratio for merging patterns, and significance cutoffs), and our implementation differs from DeepSTARR’s original pipeline. As a result, we observe both overlapping motifs, where L2G and DeepSTARR agree, and unique motifs, where one method finds patterns the other does not. A “missed” motif may simply not form a sufficiently large cluster under our TF-MoDISco settings or fall below TOMTOM’s significance threshold, rather than reflecting an absence of signal in L2G.
>
> The key takeaway is that L2G not only achieves a high PCC on this task, but also supports the same motif‐discovery analyses as DeepSTARR, uncovering relevant motifs that compliment DeepSTARR’s original findings.

---

> ### Author Response · Authors · 2025-06-26
> **Author Response (Cont.)**
>
> **Q: Section 3.2 relies a lot on the NT benchmarks as being suitable for differentiating among models. I'm not sure the benchmark has aged well, and would prefer to see more thoughtfully constructed benchmarks like BEND or DART-eval used here.**
>
> R: We focused on the NT benchmark because, at the time we developed L2G, it was one of the only few available benchmarks for genomics and had been published in a high-impact venue (Nature Methods). However, we recognize the value of newer benchmarks. Although we were unable to complete all tasks on these benchmarks within the short rebuttal period, we have evaluated L2G on the cell-type-specific element classification task from the DART-Eval benchmark. We report both overall accuracy across all classes and accuracy for each cell line. Notably, L2G is the only fine-tuned method that outperforms the non-pretrained supervised model ChromBPNet in overall accuracy.
>
> | Model                 | Overall | GM12878 | H1ESC | HEPG2 | IMR90 | K562  |
> |-----------------------|---------|---------|-------|--------|--------|-------|
> | ChromBPNet-like       | 66.7    | **88.7**    | **88.6**  | 87.4   | 87.4   | 86.5  |
> | DNABERT-2             | 65.0    | 87.8    | 88.6  | 87.5   | 86.3   | 83.6  |
> | GENA-LM               | 63.6    | 86.9    | 88.1  | 87.4   | 85.8   | 83.6  |
> | HyenaDNA              | 61.0    | 86.6    | 86.8  | 86.2   | 85.3   | 83.8  |
> | Nucleotide Transformer| 63.2    | 96.8    | 88.1  | 87.1   | 85.9   | 83.6  |
> | L2G                   | **69.0**    | 87.9    | 87.6  | **87.8**   | **87.7**   | **87.1**  |
>
> ---
>
> **Q: I think it's true that L2G's surprising performance raises the question of whether self supervised learning on large transformers is an effective way to build DNA lms, but I honestly do not think the rest of the paper pursues this question. What it interrogates is how to correct the failure of ORCA to perform well on adapting LLMs to tasks involving genomic DNA.**
>
> R: Thank you for this comment. We agree that the main focus of our paper is on how to more effectively adapt LLMs to genomics, rather than critiquing DNA LLMs themselves. However, we believe our findings highlight the need to rethink how DNA LLMs can better exploit the massive amounts of unsupervised data they are trained on.
>
> ---
>
> **Q: The embedder pre-training loss objective is described a bit vaguely. What do the authors mean by saying they 'schedule' the weighted losses? Could this concluding part of 2.4.2 be clarified?**
>
> R: We have updated the manuscript to clarify this. Specifically, we set $\alpha = 0$ during the first two thirds of the total epochs of pre-training, effectively optimizing only the task-specific loss ($L_{\text{task}}$) in the early stage. Afterward, we set $\alpha$ to a positive value and optimize the full joint objective. This staged approach empirically leads to better performance, as delaying the alignment loss allows the embedder to first learn meaningful task-relevant representations before enforcing modality alignment via $L_{\text{MMD}}$.
>
> ---
>
> **Q: At the beginning of section 3.1, the authors say that Directly applying transformer models trained on natural language data ... can lead to the corruption of pre-trained weights 'corruption' of the pre-trained weights is a strange phrase here. I think the authors mean degraded performance on the source domain, with no guarantee of good performance in the target domain?**
>
> R: We thank the reviewer for this suggestion. We have rephrased the sentence as “Directly applying transformer models trained on natural language data to out-of-domain tasks like genomics can degrade the quality of the pre-trained weights”

---

### Review · Reviewer_eAGp · 2025-05-03

**Summary Of Contributions:**

The work introduces the L2G -- Language-to-Genome -- framework that  adapts pre-trained language models to genomic prediction tasks, leveraging a carefully designed pipeline of neural architecture search. a domain-adaptation-style pretraining of a DNA embedder to close the modality gap between DNA and language modalities, and finally, task-specific supervised fine-tuning. The authors empirically show the advantages of L2G in data and computational efficiency, along with performance that is competitive with more expensive genomic foundation models. They further study the impact of different choices on performance through ablation studies. This line of work is especially useful to allow practitioners to adapt and use high performing foundation/language models under resource constraints.

**Audience:**

Yes

**Broader Impact Concerns:**

The authors sufficiently address broader impact and limitations in the Discussion section of the work.

**Claims And Evidence:**

Yes

**Requested Changes:**

As discussed above, some discussion and/or empirical evidence about the following questions would be appreciated:

- Why is a CNN a natural choice for DNA embeddings, especially as opposed to other architectures designed for sequential modeling?
- How sensitive is the method to choices of hyperparameters and pretraining data?

**Strengths And Weaknesses:**

Strengths

- The paper identifies limitations of previous works (e.g. architecture design, difficulties in cross-modality predictions, computational expense) to motivate the proposed framework. The authors address each limitation using existing techniques in neural architecture search and domain adaptation to build a 3-stage training pipeline.
- The empirical results demonstrate efficiency and effectiveness of the L2G framework. The ablation studies tease apart the impact of difference design decisions.
- They also show a meaningful biological application in discovering functional regulatory syntax, highlighting the relevance of the method in real tasks.

Weaknesses

- The motivation behind the choice for a CNN architecture for DNA embeddings was not clear, given that the data is sequential in nature, and might be more amenable to sequential modeling.
- Discussion about important hyperparameters (e.g. weights on different loss terms) or the choice of pretraining data is lacking.
- Other limitations, like limited complexity of the benchmarked tasks and lack of interpretability, are already discussed by the paper.

---

> ### Author Response · Authors · 2025-06-26
> **Author Response**
>
> **Q: The motivation behind the choice for a CNN architecture for DNA embeddings was not clear, given that the data is sequential in nature, and might be more amenable to sequential modeling.Why is a CNN a natural choice for DNA embeddings, especially as opposed to other architectures designed for sequential modeling?**
>
> R: Thank you for raising this point. We chose CNN as the backbone for DNA embeddings because CNNs are in fact well-suited for DNA (and RNA) embeddings as evident in previous work. Landmark models like DeepBind [1] and DeepSEA [2] used shallow CNNs to predict protein binding and chromatin features, and they remain standard baselines in the computational genomics field. More recent architectures (e.g., Akita [3], Basenji [4], DeepSTARR [5], ChromBPNet [6]) build on this foundation by incorporating dilated convolutions or residual connections to capture longer-range dependencies, but the core use of convolutions remains central. CNNs excel at learning local patterns, and in genomic data, they can effectively identify short regulatory patterns, such as transcription factor binding motifs, splice signals, repeat elements, etc., regardless of their position within a long genomic sequence. Each filter can recognize different DNA patterns across the sequence. We have added these clarifications in the manuscript.
>
> **References**
>
> [1] Alipanahi, Babak, et al. "Predicting the sequence specificities of DNA-and RNA-binding proteins by deep learning." *Nature biotechnology* 33.8 (2015): 831-838.
>
> [2] Zhou, Jian, and Olga G. Troyanskaya. "Predicting effects of noncoding variants with deep learning–based sequence model." *Nature methods* 12.10 (2015): 931-934.
>
> [3] Fudenberg, Geoff, David R. Kelley, and Katherine S. Pollard. "Predicting 3D genome folding from DNA sequence with Akita." *Nature methods* 17.11 (2020): 1111-1117.
>
> [4] Kelley, David R., et al. "Sequential regulatory activity prediction across chromosomes with convolutional neural networks." *Genome research* 28.5 (2018): 739-750.
>
> [5] de Almeida, Bernardo P., et al. "DeepSTARR predicts enhancer activity from DNA sequence and enables the de novo design of synthetic enhancers." *Nature genetics* 54.5 (2022): 613-624.
>
> [6] Pampari, Anusri, et al. "ChromBPNet: bias factorized, base-resolution deep learning models of chromatin accessibility reveal cis-regulatory sequence syntax, transcription factor footprints and regulatory variants." *BioRxiv* (2025): 2024-12.
>
> ---
>
> **Q: Discussion about important hyperparameters (e.g. weights on different loss terms) or the choice of pretraining data is lacking.
> How sensitive is the method to choices of hyperparameters and pretraining data?**
>
> R: We thank the reviewer for this question and have conducted additional experiments on key hyperparameters such as different loss weights and pretraining data choices.
>
> We performed ablation studies on the loss term weights (α is for alignment loss and β is for task loss).
>
> H3:
>
> | α    | β    | MCC / F1 |
> |------|------|----------|
> | 0    | 1    | 79.5     |
> | 0.1  | 1    | 79.3     |
> | 1    | 1    | 82.5     |
> | 10   | 1    | 81.7     |
> | 1    | 0    | 69.0     |
>
> Promoters TATA:
>
> | α    | β    | MCC / F1 |
> |------|------|----------|
> | 0    | 1    | 86.2     |
> | 0.1  | 1    | 96.4     |
> | 1    | 1    | 96.0     |
> | 10   | 1    | 94.5     |
> | 1    | 0    | 89.8     |
>
> Enhancers:
>
> | α    | β    | MCC / F1 |
> |------|------|----------|
> | 0    | 1    | 54.5     |
> | 0.1  | 1    | 58.1     |
> | 1    | 1    | 55.8     |
> | 10   | 1    | 57.5     |
> | 1    | 0    | 52.4     |
>
>
> We also evaluated the impact of language source data by comparing the CoNLL-2003 and JNLPBA datasets. We chose JNLPBA because it is specifically designed for biomedical named entity recognition, making it more domain-aligned with our genomic sequence analysis tasks than CoNLL-2003.
>
> | Dataset        | CONLL2003 | JNLPBA |
> |----------------|-----------|--------|
> | H3             | **82.5**      | 82.2   |
> | Promoters TATA | 96.0      | **96.2**   |
> | Enhancers      | 55.8      | **56.5**  |
>
> We found that setting β=0 consistently yields the worst performance across all tasks, highlighting the importance of the task loss term. The optimal value of α varies by tasks. As for the pretraining data choice, we observed no significant performance difference between using CONLL-2003 and JNLPBA as the source proxy dataset.

---

### Review · Reviewer_kD7P · 2025-06-12

**Summary Of Contributions:**

This paper proposes L2G, a technique for cross-modal transfer learning from language to genomic data. The technique is based on ORCA, but has suitable adaptations that better enables cross-modal learning from language to genomic data.
The paper provides a comparison to pre-trained (and fine-tuned) genomic models on three benchmarks consisting of a varied set of sub-tasks, demonstrating good performance. Further, some ablations and demonstrations of model capabilities are provided.

**Audience:**

Yes

**Broader Impact Concerns:**

None.

**Claims And Evidence:**

Yes

**Requested Changes:**

The paper would primarily benefit from expanding the ablations to increasingly demonstrate how the proposed cross-modal learning improves upon regular fine-tuning. Examples include ablations on the LM backbone, whether there are (qualitative) differences between embeddings of L2G vs. pure genomic models (expanding on Fig 7) etc.

It would also be interesting and useful if the authors could describe intuitions of why language pre-training + cross-modal transfer learning seems to lead to better results than pre-training on genomic data + fine-tuning on genomic  data (e.g. how the proposed L2G setup leverages the LM logic in a manner beneficial for the downstream genomic tasks), although this request is admittedly more vague.

Some minor typos: "an unseen modalities" (page 3), "Bridge this gap often" (page 3), "[...] to learn f into map h into [...]" (page 5). In Table 5, the second HyenaDNA accuracy is bolded when it shouldn’t be (please also consider using bold for equal top accuracies).

**Strengths And Weaknesses:**

Strengths:

The paper demonstrates that it is possible to efficiently learn genomic models from large language models using the proposed L2G transfer learning technique. On the included benchmarks, the L2G model performs better than similarly-sized models trained on considerably more genomic data. The benchmarking is thorough, and exploration of model properties are included. It is a clearly written paper with figures that generally presents their setup and findings well.

The proposed adaptations of the ORCA setup are sensible and well-justified. This includes introducing a task-specific loss to enable cross-modal transfer learning, and replacing the OTDD loss from the ORCA setup to MMD, which seemingly allows for better imitation of the reasoning of the base language model.

Weaknesses:

The paper presents some ablations on the setup, but more comparisons of design choices would be desirable. Some general examples include 1. How does the choice of the RoBERTa-base model impact performance compared to similar (or larger) language models? 2. Usage of DASH: On the shown tasks in Fig. 5, UNet embedder architecture consistently performs better than ResNet. Further, on 2 of 3 shown tasks, UNet also performed better than using DASH. It would be interesting to see a comparison of which CNN backbones (ResNet vs UNet) is selected for different tasks by DASH, and some additional results that indicate how using DASH improves over the base UNet.

The paper would also benefit from some general clarifications on some technical differences between embedder pre-training, which also seems to include the task loss and hence the predictor, and the final fine-tuning (such as the data used for pretraining vs fine-tuning, and whether the task loss for pretraining is tied to the task the model is fine-tuned on). Figure 2 part C does not sufficiently clarify this for me.

---

> ### Author Response · Authors · 2025-06-26
> **Author Response**
>
> We thank the reviewer for the feedback. Below we address the concerns and comments raised.
>
> **Q: How does the choice of the RoBERTa-base model impact performance compared to similar (or larger) language models?**
>
> R: We added an ablation comparing different LM backbones, RoBERTa-Base (149M parameters) and RoBERTa-Large (387M parameters), and report the results below:
>
> | Dataset        | RoBERTa-Base | RoBERTa-Large |
> |----------------|--------------|---------------|
> | H3             | 82.5         | **85.5**          |
> | Promoters TATA | 96.0         | **96.2**          |
> | Enhancers      | 55.8         | **57.1**          |
>
> Across all three tasks, RoBERTa-Large outperforms RoBERTa-Base, demonstrating that scaling up the model size of the transformer backbone can enhance downstream task performances.
>
> ---
>
> **Q: Usage of DASH: On the shown tasks in Fig. 5, UNet embedder architecture consistently performs better than ResNet. Further, on 2 of 3 shown tasks, UNet also performed better than using DASH. It would be interesting to see a comparison of which CNN backbones (ResNet vs UNet) is selected for different tasks by DASH, and some additional results that indicate how using DASH improves over the base UNet. ”**
>
> R: We included a table summarizing which backbones were selected for which tasks.
> | Dataset                  | Selected Model |
> |--------------------------|----------------|
> | H3                       | UNet           |
> | H3K4me1                  | ResNet         |
> | H3K4me2                  | ResNet         |
> | H3K4me3                  | ResNet         |
> | H3K9ac                   | ResNet         |
> | H3K14ac                  | ResNet         |
> | H3K36me3                 | ResNet         |
> | H3K79me3                 | ResNet         |
> | H4                       | UNet           |
> | H4ac                     | UNet         |
> | enhancer                 | UNet           |
> | enhancer_types           | UNet           |
> | promoter_all             | ResNet         |
> | promoter_tata            | UNet           |
> | promoter_non_tata        | UNet           |
> | Splice sites all         | UNet           |
> | Splice sites acceptors   | UNet           |
> | Splice sites donors      | UNet           |
>
>
>
> We have also added ablation results comparing DASH to the base UNet architecture on two additional datasets, in addition to the three tasks (H3, Enhancers, Promoters TATA) presented in the paper:
>
> | Dataset            | DASH | Base UNet |
> |--------------------|------|-----------|
> | H4                 | **78.8** | 76.7      |
> | Promoters No TATA  | **97.2** | 96.2      |
>
> DASH generally outperformed the base UNet across these tasks. This supports our design motivation: while base UNet might perform well, DASH provides flexibility to adaptively select the best-performing architecture per task, which can lead to further improvements over using a fixed architecture.

---

> ### Author Response · Authors · 2025-06-26
> **Author Response (Cont.)**
>
> **Q: The paper would also benefit from some general clarifications on some technical differences between embedder pre-training, which also seems to include the task loss and hence the predictor, and the final fine-tuning (such as the data used for pretraining vs fine-tuning, and whether the task loss for pretraining is tied to the task the model is fine-tuned on). Figure 2 part C does not sufficiently clarify this for me.**
>
> R: We thank the reviewer for this suggestion and have added additional clarifications to the manuscript.
> During embedder pretraining, the encoder is trained with a joint objective that combines both a task-specific loss and an alignment loss. Specifically, the embedder is paired with a linear prediction head and optimized to perform the same downstream task as in the final fine-tuning stage. The task-specific loss is computed using the same genomic dataset that is later used for fine-tuning. The alignment loss is computed using both the genomic dataset and an external language dataset (CoNLL-2003), encouraging the embedder to learn cross-domain representations. After the embedder pretraining, we retain only the encoder, attach it to the transformer backbone and a new linear predictor, and then perform final fine-tuning.
>
> **Q: It would also be interesting and useful if the authors could describe intuitions of why language pre-training + cross-modal transfer learning seems to lead to better results than pre-training on genomic data + fine-tuning on genomic data (e.g. how the proposed L2G setup leverages the LM logic in a manner beneficial for the downstream genomic tasks), although this request is admittedly more vague.**
>
> R:
> The intuition behind why language models combined with cross-modal transfer can outperform in-domain DNA LLMs is two-fold:
>
> First, while genomic foundation models are pretrained on large-scale genomic data, they often directly borrow architectures and training recipes from NLP without tailoring them to the unique characteristics of genomic sequences. Recent studies (e.g., [1][2][3]) have shown that such pretrained models can sometimes underperform randomly initialized models on downstream tasks or simple CNN baselines. Our findings further support this observation.
>
> Second, although cross-modal transfer learning might appear counterintuitive, it has proven effective in other domains, such as computer vision [4] and PDE solving [5]. One major reason is the scale and optimization invested in general-purpose language models—efforts behind models like BERT, GPT, and RoBERTa far surpass those in domain-specific models. Our L2G framework leverages the generalization capacity and “reasoning logic” embedded in these LLMs and aligns them with genomic representations through supervised alignment. This results in an embedder that is not only initialized with rich prior knowledge from language but also regularized by genomic supervision, leading to better downstream performance.
> This alignment-then-fine-tuning strategy mirrors findings from the ORCA paper, where showed that such training can effectively harness shared knowledge across modalities.
>
> **References**
>
> [1] Tang, Ziqi, et al. "Evaluating the representational power of pre-trained DNA language models for regulatory genomics." *bioRxiv* (2024).
>
> [2] Xu, Zongzhe, et al. "Specialized foundation models struggle to beat supervised baselines." *arXiv preprint* arXiv:2411.02796 (2024).
>
> [3] Vishniakov, Kirill, et al. "Genomic foundationless models: Pretraining does not promise performance." *bioRxiv* (2024): 2024-12.
>
> [4] Lu, Kevin, et al. "Frozen pretrained transformers as universal computation engines." *Proceedings of the AAAI conference on artificial intelligence.* Vol. 36. No. 7. 2022.
>
> [5] Shen, Junhong, Tanya Marwah, and Ameet Talwalkar. "Ups: Efficiently building foundation models for pde solving via cross-modal adaptation." *arXiv preprint* arXiv:2403.07187 (2024).

---

### Decision · Action_Editor_pdVe · 2025-07-15

**Recommendation:** Accept as is

**Audience:**

Yes

**Audience Explanation:**

This paper will be of interest both for the audience working on genomic foundation models as well the audience working on fine-tuning methodology.

**Claims And Evidence:**

Yes

**Claims Explanation:**

The authors have been very dedicated to support their claims with numerical evaluations.